# Predictive Biomarkers of Methotrexate Treatment Response in Patients with Rheumatoid Arthritis: A Systematic Review

**DOI:** 10.3390/metabo15110715

**Published:** 2025-10-31

**Authors:** Adla B. Hassan, Rowida M. Hamid, Saja H. Alamien, Namaa A. Khalil, Duaij Salman Saif, Mohammed Elfaki, Haitham Jahrami

**Affiliations:** 1Department of Internal Medicine, College of Medicine and Health Sciences, Arabian Gulf University, Manama 26671, Bahrain; 2Department of Pharmacotherapy, National University, Khartoum, Sudan; rowida.mohammed83@gmail.com; 3College of Medicine and Health Sciences, Arabian Gulf University, Manama 26671, Bahrain; sajahassan143@gmail.com (S.H.A.); duaijsaif2323@gmail.com (D.S.S.); haitham.jahrami@outlook.com (H.J.); 4Medical Skills and Simulation Center (MSSC), College of Medicine and Health Sciences, Arabian Gulf University, Manama 26671, Bahrain; dr.namaakhalil@gmail.com (N.A.K.); elfakymo@gmail.com (M.E.); 5Government Hospitals, Manama 329, Bahrain

**Keywords:** MTX, RA, metabolomics, metabolites, DMARD, clinical DAS28 score

## Abstract

**Background:** Methotrexate (MTX) is the most used anti-rheumatic drug for the treatment of early rheumatoid arthritis (ERA) patients, with an adequate response rate of only 30–40%. Thus, early detection of response failure is very crucial to prevent permanent disability. **Objectives:** We aimed to provide an update on the current evidence of potential predictive biomarkers of MTX treatment response (MTX-TR) in patients with ERA. **Materials and Methods:** PubMed/MEDLINE, Scopus, EBSCO, and Cochrane Library were searched for studies that investigated a multitude of predictive metabolites of MTX-TR in ERA patients during the 2000–2024 period. This study was registered in PROSPERO (ID: CRD42024547651). **Results:** We determined that 31 out of 102 metabolites studied were the best predictive of MTX-TR in ERA, using clinical response (DAS28-ESR score). Our results on serum protein profiles revealed that higher pre-treatment levels of myeloid-related proteins, MTX–polyglutamates, choline, inosine, hypoxanthine, guanosine, nicotinamide, and diglyceride, and lower pre-treatment levels of N-methyl isoleucine, 2,3-dihydroxy butanoic acid, nor-nicotine, glucosylceramide, and itaconic acid, were associated with a good MTX-TR. However, lower baseline plasma itaconate and its derivatives and haptoglobin, but a higher baseline level of galactosylated glycans (FA2G) of IgG1, were associated with a good response to MTX. The results on immune cell biology indicated that higher pre-treatment of regulatory B cells, lower pre-treatment of Treg, and RDW were correlated with a good MTX-TR. The results on inflammatory biomarkers showed that a lower IL-1ra/IL1B ratio and IL-6 levels after MTX indicated a good response. **Conclusions:** This study provides an update on the current evidence of the potential predictive metabolites for the best MTX-TR in ERA patients. We revealed that few biomarkers resulted in a remission state of patients with ERA. These biomarkers are promising but not yet ready for routine clinical use; they warrant validation in larger prospective trials. We recommend that, for the implementation of personalized medicine, these biomarkers should be the first-line biomarkers for use in routine clinical practice after validation.

## 1. Introduction

Rheumatoid arthritis (RA) is one of the rheumatic and musculoskeletal diseases (RMDs). It is characterized by chronic inflammation, joint destruction, and deformity that significantly affects joint mobility and function. Thus, patients with RA have a reduced quality of life with a high rate of morbidity [1] and mortality [2,3]. The existence of anti-cyclic citrullinated peptide (anti-CCP) and rheumatoid factor (RF) antibodies, either together or one of them, in any patient with RA would classify the disease as seropositive RA, while their absence would classify it as seronegative RA. Early RA (ERA) can be subdivided into three clinical phenotypes based on a distinct pathophysiology feature, prognosis, or outcome. The three clinical phenotypes were named undifferentiated arthritis (UA), seropositive RA, and seronegative RA. Baseline radiographic erosions, or the existence of anti-CCP and/or RF, were associated with a more severe and aggressive phenotype of RA disease [4,5,6]; thus, it needs more intensive treatment compared to seronegative RA [7,8]. These features are prerequisites for personalized medicine.

Methotrexate (MTX) is the most used conventional synthetic disease-modifying anti-rheumatic drug (csDMARD) for the treatment of patients with RA based on its safety, efficacy, and cost [9]. Early treatment with MTX plus glucocorticoids and, subsequently, other csDMARDs or biological DMARDs prevents RA-related disability in up to 90% of patients and improves outcomes [10]. Old and recent clinical studies have constantly demonstrated the benefits of MTX use in patients with RA in reducing disease activity, joint damage, and radiographic progression [11,12]. MTX exerts strong anti-inflammatory and immunomodulatory effects that diverge from its antiproliferative action observed in oncology. Its therapeutic action is primarily mediated through two major mechanisms: folate antagonism and adenosine-mediated anti-inflammatory signaling [13,14,15,16]. MTX competitively inhibits dihydrofolate reductase (DHFR) and thymidylate synthase, which are critical enzymes for tetrahydrofolate (THF) regeneration and purine and pyrimidine synthesis. This inhibition suppresses DNA synthesis and cell proliferation, particularly impacting activated T and B lymphocytes. Additionally, intracellular MTX–polyglutamates inhibit AICAR (5-aminoimidazole-4-carboxamide ribonucleotide) transformylase, causing the accumulation of AICAR, a purine biosynthesis precursor [14]. Collectively, these actions diminish the activation and proliferation of immune cells. Regarding adenosine signaling, the accumulation of AICAR from MTX inhibits adenosine deaminase and AMP deaminase, consequently raising extracellular adenosine levels. Adenosine induces potent anti-inflammatory effects by binding to A2A and A3 receptors on immune and endothelial cells, subsequently suppressing pro-inflammatory cytokines (TNF-α, IL-6, IL-8) and inhibiting leukocyte adhesion and migration [15,16,17]. This pathway is now recognized as the principal anti-inflammatory mechanism underpinning low-dose MTX therapy in RA. Blockade of adenosine receptors or disruption of extracellular adenosine formation negates MTX’s anti-inflammatory efficacy in experimental models, reinforcing the significance of this mechanism [16]. The interplay between these pathways connects MTX’s pharmacological effects to metabolic processes, making it a prime candidate for metabolomic investigations in RA. MTX treatment has been shown to modify purine metabolism, amino acid turnover, and oxidative stress markers, indicating these alterations as potential biomarkers for treatment response, particularly in early rheumatoid arthritis [18].

On the other hand, MTX is known to be associated with possible side effects, such as bone marrow suppression and hepatotoxicity [19]. Although MTX has improved the inflammatory and destructive course of the disease in many cases, a significant percentage (60–70%) of RA patients still do not respond to an adequate level of MTX in the first 6 months of its initiation [20]. Predictors of MTX treatment success or failure for individual patients have not been well defined yet. The lack of such predictors prevents the early initiation of such effective drug therapy.

Recently, it was shown that RA disease is accompanied by metabolic alterations resulting in huge metabolomic profiles. These metabolomic profiles can be determined using targeted and non-targeted metabolomics technology [21]. To date, a few studies have investigated the metabolic changes associated with disease activity in RA. Metabolomic profiles have been used to identify predictive biomarkers of MTX response in RA patients [22,23]. RA patients’ metabolomic profiling revealed the metabolic signatures that differentiate them from healthy controls. The metabolic signatures emphasize the disruptions in amino acid metabolism, lipid metabolism, energy production pathways, inflammatory injury, oxidative stress, and phospholipid metabolism [18,24]. For instance, in RA patients, elevated levels of branched-chain amino acids and alterations in the tricarboxylic acid cycle have suggested a link between disease pathology and metabolic dysregulation [24]. Further studies have identified specific protein metabolites, such as tryptophan and L-leucine, as potential biomarkers for treatment response and reduced disease activity in RA [18]. Furthermore, some studies highlighted the effects of the intestinal microbiota/microbiome on RA disease, specifically on arthritis development [25], pathogenesis [26], and the pharmacological activity of MTX treatment [27]. In 2025, a recent study analyzed dried blood spot samples from RA patients and healthy controls. It showed that six key metabolites linked to fatty acid oxidation and amino acid metabolism could serve as an effective diagnostic biomarker for RA [28]. Another review study showed that disturbances in energy, lipid, and amino acid metabolism across three different body fluids from RA patients provided critical information about the metabolic profile related to drug response, disease activity, and comorbidities [29]. Regarding the clinical response to MTX, it has been shown that a high baseline (before MTX treatment) M-ficolin plasma level strongly correlated with high disease activity. Therefore, in early RA patients, the reduced plasma level of M-ficolin after treatment with MTX, intra-articular glucocorticoids, and biologics was considered a strong predictor of remission and low disease activity, as assessed by DAS28 (disease activity score in 28 joints) [30]. Regarding genetic studies, in 2024, two recent studies showed that single nucleotide polymorphisms in genes governing the MTX cellular pathways, including the folic acid and adenosine pathways, are associated with MTX efficacy and toxicity [31,32].

To highlight the unique contribution of this systematic review, Table 1 compares our study with existing systematic and scoping reviews, emphasizing its exclusive focus on methotrexate-specific predictive biomarkers in early rheumatoid arthritis (see Table 1).

To date, the lack of reliable and definitive predictive biomarkers regarding the outcomes of patients undergoing MTX treatment leads, in many cases, to medical errors. To enable quicker therapy adjustments, the earlier identification of responders and non-responders to MTX will be of great clinical importance in personalized medicine. Therefore, the primary aim of this systematic review was to provide an update on the current data regarding potential biomarkers that predict the best clinical response to MTX treatment in early RA patients. Also, we aimed to investigate if there are any first-line biomarkers for use in routine clinical practice. A secondary aim was to identify what characterizes responders and non-responders to guide MTX drug therapy and to serve as a foundation for precision medicine in early RA patients.

## 2. Materials and Methods

PubMed/MEDLINE, Scopus, EBSCO, and Cochrane Library were searched for randomized controlled trials (RCTs) and original or observational studies that investigated a multitude of potential biomarkers and predictors of MTX treatment response (MTX-TR) in patients with early RA during the 2000–2024 period. We also manually checked the references of the included articles for potential inclusion in this systematic review. This search resulted in the identification of several eligible studies. The MTX-TR is mainly based on the changes in DAS28 scores. The patients included were patients with definite RA according to American College of Rheumatology (ACR) criteria or European League Against Rheumatism (EULAR) or 2010 ACR/EULAR criteria [33,34]. The current study was registered in PROSPERO in June 2024; registration ID: CRD42024547651. The PRISMA 2020 checklist is available in the Appendix A.

DAS28-ESR is a tool for the assessment of disease activity. It is calculated from a patient’s global assessment (0–100 mm visual analog scale), swollen joint, and tender joint counts (SJC; TJC) out of the 28 joints and erythrocyte sedimentation rate (ESR, mm/hr) [34]. Of note, one important point when using the DAS28 score (0–10) to assess the response to treatment, a DAS-28 reduction of 0.6 after therapy represents a moderate improvement, while a reduction of more than 1.2 represents a major or significant improvement [9]. Another important point regarding DAS28 is that a score of <2.6 suggests disease remission, 2.6–3.2 suggests low disease activity, >3.2–5.1 suggests moderate disease activity, and >5.1 suggests high disease activity.

Using the PICO framework, our research question was as follows: In patients with early rheumatoid arthritis who are naïve to MTX therapy (P), how do baseline metabolomic biomarkers change after MTX treatment (I), compared to standard clinical measures alone (C), and predict the therapeutic response to methotrexate (O), as measured by DAS28-ESR scores. The primary effect measure used in this study was the change in DAS28-ESR score from baseline following MTX treatment, which was used to determine the clinical response (responder vs. non-responder classification).

In all studies, clinical data were collected from patients prior to the initiation of MTX and again at the end of the study duration. The eligibility for participation in this study included the following: ≥18 years of age, fulfillment of criteria for the diagnosis of rheumatoid arthritis as mentioned above, and no prior exposure to MTX treatment. The inclusion criteria for the studies included the following: they should include only rheumatoid arthritis patients with an early diagnosis of 6–12 months, they should be human studies, they should consider methotrexate treatment and its efficacy as a first-line therapy, and they should investigate metabolomes/metabolite/metabolic biomarkers and their influence on MTX treatment. The studies should be only RCTs or observational studies. Studies that involved participants of unisex or lacked a control or placebo group were not excluded. The exclusion criteria included the following: non-rheumatoid arthritis patients/did not use methotrexate, languages other than English, animal models, abstracts only, notes, expert opinion, letters, reviews, systematic reviews or meta-analyses, and case reports with fewer than 3 cases.

Using the two basic Boolean operators “AND, OR”, four different search strategies were designed and used to search the four databases included in the current study as follows:

1. (((“Metabolome”[Mesh] OR Metabolome[tiab] OR metabolite[tiab] OR Nmethylisoleucine[tiab] OR Nornicotine[tiab] OR “Biomarkers”[Mesh:NoExp] OR “bio-markers”[tiab]) AND (“Methotrexate”[Mesh] OR Methotrexate[tiab] OR MTX[tiab] OR Rheumatrex[tiab] OR Trexall[tiab] OR Otrexup[tiab])) AND (“Treatment Outcome”[Mesh] OR Efficacy[tiab] OR Effectiveness[tiab] OR Response[tiab])) AND (“Arthritis, Rheumatoid”[Mesh] OR “Rheumatoid arthritis”[tiab] OR RA[tiab].

2. (((“Metabolome”[Mesh] OR Metabolome[tiab] OR metabolite[tiab] OR Nmethylisoleucine[tiab] OR Nornicotine[tiab] OR “Biomarkers”[Mesh:NoExp] OR “biomarkers”[tiab]) AND (“Methotrexate”[Mesh] OR Methotrexate[tiab] OR MTX[tiab] OR Rheumatrex[tiab] OR Trexall[tiab] OR Otrexup[tiab])) AND (“Arthritis, Rheumatoid”[Mesh] OR “Rheumatoid arthritis”[tiab] OR RA[tiab])) AND (“Treatment Outcome”[Mesh] OR Efficacy[tiab] OR Effectiveness[tiab] OR Response[tiab]) AND DAS-28[tiab].

3. metabol* OR biomarkers in Title Abstract Keyword AND Methotrexate OR MTX OR Rheumatrex OR Trexall OR Otrexup in Title Abstract Keyword AND “rheumatoid arthritis” in Title Abstract Keyword AND “efficacy” OR response OR “treatment outcome” in Title Abstract Keyword AND “DAS28” in Title Abstract Keyword.

4. (TITLE-ABS-KEY (metabolome OR metabolite OR nmethylisoleucine OR nornicotine OR das-28) AND TITLE-ABS-KEY (methotrexate OR mtx OR rheumatrex OR trexall OR otrexup) AND TITLE-ABS-KEY (“rheumatoid arthritis” OR ra)) AND (rheumatoid AND arthritis) AND (methotrexate) AND (biomarker) AND (metabolite) AND (metabolome) AND (LIMIT-TO (DOCTYPE, “ar”)) AND (LIMIT-TO (LANGUAGE, “english”).

“Among the sixteen included studies, three were randomized clinical trials (RCTs) and thirteen were cohort studies. Accordingly, two tools were employed to assess the risk of bias (ROB): the Cochrane ROB tool for the randomized trials and the PROBAST tool for the cohort studies. Four reviewers independently assessed the ROB; disagreements were resolved by discussion with a fifth reviewer.

Regarding the Cochrane ROB tool (domains: random sequence generation, allocation concealment, blinding of participants/personnel, blinding of outcome assessment, incomplete outcome data, selective reporting, and other bias), each domain was judged as “low”, “unclear”, or “high” risk. An overall ROB judgment was derived as follows: studies were considered “low risk” if all domains were rated low, “high risk” if any domain was high, and “unclear” if no domain was high but at least one was unclear. Our results revealed that one study showed high ROB, and two studies showed unclear ROB.

On the other hand, the PROBAST framework evaluates ROB across four key domains: participants, predictors, outcome, and analysis. Following the PROBAST guidance, each domain was rated as “low,” “high,” or “some concerns,” with the overall ROB judgment determined by the highest risk rating across all domains. Out of the 13 studies, 11 studies (84.6%) were judged to have an overall high ROB, while 2 studies (15.4%) were assessed as having some concerns regarding the overall ROB. The predominant source of bias was in the analysis domain, where most studies failed to perform appropriate statistical steps essential for prediction modeling, such as internal or external model validation.”

Regarding statistical analysis, in most studies, the Wilcoxon test for paired samples was used to compare pre- and post-treatment data. Some studies did not report frequency or association; therefore, the statistical significance was determined and compared with the direct results of the studies as percentages.

## 3. Results

A total of 534 articles were retrieved from the four databases: PubMed/MEDLINE and PMC (207), Scopus (31), EBSCO (58), and Cochrane Library (238). Forty-six articles were excluded as duplicates. After applying the eligibility criteria to the titles, abstracts, and full articles, another 472 studies were excluded. A total of 16 studies were included for the final analysis; the results are depicted in a PRISMA 2020 flowchart, as shown in Figure 1. The selection of the relevant articles was performed and reviewed by five independent doctors. The 16 included studies were found to be from 11 different countries (Austria, Denmark, China, Japan, Spain, USA, Switzerland, India, Sweden, Russian and Italy). As a result of a detailed and comprehensive final analysis of the 16 articles included in the current study, the total number of biomarkers investigated was found to be 102 (Table 2). Those 102 biomarkers included metabolomics, immune cell biology, inflammatory pathways, and serum protein profiles. The total number of patients included in this review was 946 patients, with a mean of 59.13 patients per study, range 19–163. The total duration of the studies in weeks (Ws) was 465 Ws (mean of 29.06, range of 12–129 Ws) (Table 2). Our results, as depicted in Table 2 and Table 3, showed that the age of the patients was in the range of 32.5–86 years. Most of the patients were females, at 78.61% (746). The results depicted in Table 3 and Figure 2 demonstrate the number of patients and responders in each study and reveal that the total percentage of MTX responders in the current review was 51% (482 patients). Not all studies used the same dose of MTX. As shown in Table 4, most of the studies used a high dose of 20 or 25 mg/week, but few studies used a medium dose that ranged from 7.5 to 15 mg/week. Our results, as presented in Table 5, reveal the outcomes of MTX treatment response (MTX-TR) according to the 102 metabolomes studied; only 31 out of those 102 metabolites were found to be functionally important and predicted the response to MTX. The patients were categorized as responders if their DAS-28 was ≤3.2 or the score decreased or changed by >1.2 from the baseline, as described previously [35], and non-responders if they did not fulfill the EULAR or ACR response criteria [33,34]. Of note, one important point when using the DAS28 score (0–10) to assess response to treatment, a DAS-28 reduction of 0.6 after therapy represents a moderate improvement, while a reduction of more than 1.2 represents a major improvement. Another important point regarding DAS28 is that a score of <2.6 suggests disease remission, 2.6–3.2 suggests low disease activity, >3.2–5.1 suggests moderate disease activity, and > 5.1 suggests high disease activity. Figure 3 shows a major reduction in the DAS28 score to a remission state in five studies, to low disease activity in three studies, and to moderate disease activity in seven studies, which are considered major improvements. Figure 3 shows that the disease activity reached remission (DAS28 < 2.6) in five studies: Daly R. et al., 2020 [36], (increased levels of itaconate and its derivatives in response to MTX treatment), Chara L. et al., 2015 [37], (the higher pre-treatment number of circulating monocytes and its three subset cells), Nishina N. et al., 2013 [38] (serum IL-6 levels significantly reduced after MTX treatment in early RA pts), Wang, Z. et al., 2012 [39] (serum levels of 11 endogenous metabolites associated with good MTX-TR), and Wolf J. et al., 2004 [40] (the absence or the presence of both functional multi-resistant protein and reduced folate carrier (fMRP and RFC), together at the same time). Figure 4 shows the metabolites characterized into three groups according to the changes in the DAS28 score after MTX treatment, including the metabolites that indicated remission, those that indicated low disease activity, and those that indicated moderate disease activity.

Table 6 summarizes the 31 predictive biomarkers of methotrexate (MTX) treatment response in early rheumatoid arthritis (ERA), categorized into metabolites, lipids, serum proteins, immunoglobulins, immune cells, and cytokines. Upregulated biomarkers in good responders, such as choline, MTX–polyglutamates, and transitional regulatory B cells, reflect effective MTX metabolism and immune regulation, while downregulated markers, like itaconate, haptoglobin, and high monocyte counts, indicate a poor response, linked to persistent inflammation. These findings provide a clinically actionable framework for predicting MTX efficacy, supporting personalized medicine in ERA.

Based on the risk of bias assessment, the included observational studies demonstrated variable methodological quality. The majority of studies (11 of 13) were rated as having an overall high risk of bias, primarily driven by concerns in the analysis domain, where issues such as inadequate confounder adjustment, missing data handling, or inappropriate statistical methods were identified. Two studies (Wolf et al., 2005 [40] and Patro et al., 2016 [47]) achieved an overall rating of “some concerns,” demonstrating better methodological rigor, particularly in their analytical approaches (Figure 5).

Based on the risk of bias assessment using the Cochrane risk of bias tool, the three included randomized controlled trials demonstrated moderate methodological quality with notable limitations. All three studies received an overall rating of either high or unclear risk of bias. The primary concerns stemmed from inadequate reporting of randomization procedures, with sequence generation being unclear in two studies (Fortea-Gordo P et al., 2020 [44] and Hansen et al., 2006 [50]) and allocation concealment being unclear in all three trials. Fortea-Gordo P et al. 2020 [44] was rated as having a high overall risk of bias due to a lack of blinding of the participants and personnel, which introduces potential performance bias. The remaining domains, including incomplete outcome data and selective reporting, were generally well-addressed across studies. The unclear risk of bias in key methodological domains, particularly regarding allocation concealment, limits confidence in the internal validity of these trials and suggests that the treatment effects may be subject to selection and performance bias (Figure 6).

## 4. Discussion

The implementation of personalized medicine in rheumatoid arthritis (RA) in daily practice is still underprivileged due to a lack of strong evidence. Hence, the identification of biomarkers for the prediction of MTX response is an important clinical issue that can assist in identifying RA patients who may respond to MTX monotherapy. In the current systematic review study, a detailed and comprehensive final analysis of the 16 articles included from 11 different countries revealed that the total number of biomarkers investigated during the period 2000–2024 was 102 biomarkers or metabolomes. Clinical data were collected from all patients prior to initiation of MTX therapy and again at the end of the study duration (with a mean of 29.06 weeks). The total number of patients included in this review was 946 adults with early RA; most of them were females, at 78.61% (746). In the current study, the evaluation of the relationship between pre-treatment plasma metabolite levels and change in the DAS28-ESR score over the treatment period resulted in discrimination between MTX responders (51%, 482 patients) and non-responders. Our results demonstrated that the 102 metabolomes reported in our study include inflammatory pathways, serum protein profiles, and immune cell biology, as described below. Furthermore, only 31 out of the 102 metabolites were found to be functionally important and predicted the response to MTX. Moreover, a few of the 31 metabolites reported by five studies were found to be the best predictors of MTX-TR as they predicted the remission state of patients with RA. These are itaconate and its derivatives, circulating monocytes and their three subset cells, serum IL-6 levels, serum levels of 11 endogenous metabolites, and functional multi-resistant protein, together with reduced folate carrier (fMRP and RFC).

### 4.1. Inflammatory Pathways

Inflammatory pathways were investigated by three studies. Seitz M and his group [51] revealed that a lower IL-1ra/IL1B ratio (to less than 100) after MTX therapy indicated a good response to MTX, which was mainly due to constitutively increased IL-1ß produced by PBMC, but constitutive TNF-α release from PBMC was not detectable in their patients [51]. Additionally, Nishina et al. [38] revealed that high baseline levels of inflammatory biomarkers, such as IL-6, CRP, and CCL19, but not TNF-α, were associated with radiographic progression in patients with early RA [38]. On the other hand, in 2006, Hansen et al. [50] showed that p-CXCL12 was not associated with MTX response. The author stated that the p-CXCL12 level was constantly high and independent of any ACR disease activity variables, as well as response to MTX treatment [50].

### 4.2. Serum Protein Profiles

In the current study, serum protein profiles were investigated by 9 studies out of the 16 studies. When we summarized those nine studies, we found that in RA patients, the lower baseline or pre-treatment plasma levels of N-methyl isoleucine, 2,3-dihydroxy butanoic acid, nor-nicotine, glucosylceramide, and itaconic acid and higher baseline or pre-treatment plasma levels of choline, inosine, hypoxanthine, guanosine, nicotinamide, and diglyceride are associated with a good response to MTX therapy [39,41]. Similarly, a higher baseline of MRP8/14 (myeloid-related protein) [47], MTX-PG1-7, especially short-chain MTXPG2 [49], MTX-PG 1-5 [46], and the absence or presence of both fMRP (functional multi-resistant protein) and RFC (reduced folate carrier) together [40] were all correlated with a good response to MTX therapy. In contrast, a higher baseline or pre-treatment of plasma itaconate, itaconate anhydrase, itaconate CoA [36], FA2 of IgG1 [45], and haptoglobin [48] were correlated with an inadequate or no response to MTX therapy.

Medcalf M. et al. studied 19 plasma metabolomes. They confirmed that lower pre-treatment plasma levels of only three metabolites (N-methyl isoleucine, 2,3-dihydroxy butanoic acid, and nor-nicotine) were associated with a greater reduction in DAS28-ESR after MTX therapy [41]. Wang Z. et al. studied 20 endogenous metabolites; among them, only 11 endogenous metabolites were correlated with MTX treatment response in patients with ERA. However, six of them were elevated and five were decreased [39]. Serum MRP8/14 levels were studied by Patro P and his group, and they reported that they were correlated with disease activity at baseline and were reduced on treatment with MTX in patients who responded to treatment. In addition to that, higher baseline MRP8/14 levels were associated with the response to MTX treatment. They also showed an association between serum levels of MRP8/14 and disease activity measured by DAS28 [47]. Their results were consistent with previous studies [52,53]. Hobl et al. [49] investigated seven biomarkers of MTX–polyglutamate (MTX-PG 1-7) and showed that short-chain MTX-PG2 was a potential biomarker for clinical outcome in RA and was positively correlated with an improvement in DAS-28 [49]. Similarly, Murosaki and his group [46] showed that MTX-PG 1-5 in erythrocytes were potential indicators and predictors of MTX efficacy. Furthermore, their results indicated that each MTX-PG concentration in erythrocytes is a better indicator of clinical response to MTX than the dose of MTX itself [46]. Our results were consistent with previous studies, which showed that the detectability of MTX-PG5 in RBCs was a biomarker for the clinical response to MTX, but most patients had undetectable levels of MTX-PG5 in RBCs [54]; moreover, the RFC1 80G > A mutation was associated with a low detectability of MTX-PG [55]. Wolf J et al. stated that the absence or presence of both the fMRP and RFC biomarkers together and not either of them alone led to a significantly better MTX therapeutic outcome in RA patients evaluated by the EULAR response criteria and the ACR 20% improvement criteria [40]. In 2020, Daly R and his team studied nine plasma metabolites, as shown in Table 4. Among all the metabolites, they showed that increased itaconate was correlated with an improved DAS44 score and decreased levels of C-reactive protein (CRP). Thus, for the first time, they revealed a link between itaconate production and resolution of inflammatory disease in humans [36], which is consistent with its anti-inflammatory role as an antimicrobial effect [56]. However, it has been shown that in patients with inflammatory polyarthritis, routine clinical and laboratory factors at presentation were poor predictors of MTX treatment outcomes [57]. On the other hand, another paper showed that blood fat measurements of patients with early stages of RA were not useful for guiding methotrexate therapy, while clinical data alone gave better predictions [58].

The authors Lundstrom S et al. investigated 19 biomarkers of immunoglobulin G1 and G2 (12 Fc IgG1 and 7 Fc IgG2). They revealed that the glycosylation status of immunoglobulin G-Fc (IgG-Fc) is changed in untreated patients with early RA. Furthermore, they confirmed that a low ratio of FA2/(FA2G1 + FA2G2) of IgG1 at baseline was significantly associated with nonresponse to MTX therapy [45]. In the present study, Tan et al. [48] showed that high serum levels of haptoglobins (Haps) at baseline are associated with an inadequate response to MTX treatment after 12 weeks in early RA patients, but he could not predict the structural damage at the 1-year follow-up [48]. It is known that high Hap levels in RA might be a consequence of the activation of multiple inflammatory pathways, and it is correlated with disease activity in RA [59]. It has been shown that its main function is to bind free hemoglobin (Hb) and to prevent the damage caused by it [60]. Our study regarding plasma metabolites is consistent with another study on patients with RA and a long-term disease duration (≥9 years), which used MTX among other biological medications. The study showed that certain metabolites, such as bilirubin and serine, were associated with low disease activity [61].

### 4.3. Immune Cell Biology

Immune cell biology was investigated by four studies. A summary of this metabolome group indicated that lower baseline or pre-treatment of regulatory T cell (Treg) [42] and Red Cell Distribution Width (RDW) [43] was correlated with a good response to MTX therapy. On the other hand, higher baseline frequencies of circulating transitional regulatory B cells (cTrB), but not MatN B cells or cMem B cells, were associated with a good response to MTX treatment [44], while higher baseline numbers of circulating monocytes and their three subset cells (CD14+high/lowCD16−/+) were associated with a reduced clinical response to MTX treatment [37]. Avdeeva A, et al. [42] investigated CD4+FoxP3+Treg cells and showed that in early untreated RA, MTX treatment increases both the proportion and absolute number of Treg, with a high level of activation markers [42]. Their results were consistent with previous studies suggesting that the baseline analysis of naïve T cells may help predict the response to MTX monotherapy [62] or in combination with a biologic agent, such as antitumor necrosis factor (anti-TNF) [63]. Similarly, Gordo et al. [44] investigated three B cell and two T cell biomarkers. Their study showed that higher pre-treatment baseline frequencies of circulating transitional regulatory (cTrB) B cells (CD19+CD24hiCD38hi), but not MatN or cMem B cells, were associated with a good EULAR response to MTX [44]. Of note, the two T cells were co-cultured with B cells to support antibody production, as it was previously reported that T/B cell interactions will support antibody and cytokine secretion [64]. Bellan et al. [43] showed that a lower pre-treatment of RDW was correlated with a good response to MTX therapy. Thus, the larger the increase from baseline after MTX initiation, the better the patient’s response to it [43]. Chara L. et al. [37] revealed that a higher pre-treatment number of circulating monocytes and higher numbers of their three subset cells, as shown in Table 4, provide good predictive biomarkers of a reduced clinical response to MTX in RA patients [37].

The prevalence of adults with RA worldwide is approximately 0.8–1%. If untreated, 20–30% of RA patients become permanently disabled within the first three years following initial diagnosis [65]. On the other hand, even if treated early with MTX, there is a limited response rate to MTX. Thus, 30–40% of RA patients could experience an adequate response to monotherapy with MTX [66,67]. Moreover, in RA patients, even after a previous failure of MTX, it is still frequently effective if reintroduced again [68]. Later, it was recommended that combination therapy of a biologic agent and MTX should be initiated without delay in patients who do not have a satisfactory response to treatment with MTX alone [69]. The current EULAR and ACR recommendations advise starting methotrexate in combination with glucocorticoids as first-line treatment in ERA [70,71]. In the present study, glucocorticoids and non-steroidal anti-inflammatory drugs were permitted in many of the studies, but in low and stable doses. Thus, glucocorticoids were allowed in all studies except four. Unfortunately, it was already reported that RA patients who used corticosteroids were more likely to develop MTX intolerance [72]. Despite the intake of prednisone and independent of the degree of clinical response, our study revealed that the total percentage of MTX responders was 51%. With such a huge number of metabolomes and patients studied in the current systematic review, we would expect a higher percentage of responders. The use of glucocorticoids could be one factor for the low percentage of responders; however, this explanation needs more clarification. Another factor affecting our results could be the route of MTX therapy. In the present study, oral MTX was initiated in all patients at a low dose weekly, and when DAS28-ESR > 2.6, the MTX dose was increased gradually until the end of the study to the dose of 20 mg weekly in most of the studies or 25 mg in only four studies. In RA patients, it has been found that subcutaneous administration of MTX was significantly more effective than oral administration of the same MTX dosage [13]. This could be one of the drawbacks or limitations of all the studies included in the current systematic review that led to a relatively low percentage of responders.

The DAS28-ESR score proved to be more effective in assessing MTX response in ERA patients than other measurement scores or tools [73]. Moreover, a recent study validated a prediction model for MTX response within 24 weeks in ERA patients, which showed that baseline DAS28-ESR, anti-CCP, and the HAQ score were the top predictors of a good response to MTX [74]. However, they studied four RCTs. The current study included three RCTs, twelve prospective cohort studies, and only one retrospective cohort study. Moreover, the therapeutic response to MTX in ERA was assessed mainly with clinical scores, including DAS28-ESR, ACR, and EULAR criteria in all studies, in addition to the structural damage score (Sharp score), functional markers (HAQ), and inflammatory markers (CRP, IL-6, TNF-alfa) in a few studies. Thus, comparing those assessment values before and after MTX treatment provides invaluable data on MTX treatment efficacy. Interestingly, our current study, as shown in Figure 4, categorizes predictive metabolites into three groups according to changes in the DAS28 score after MTX treatment. Those three groups include the following: metabolites that indicated remission, those that indicated low disease activity, and those that indicated moderate disease activity.

The other predictive biomarkers of the therapeutic response in RA patients include demographic data, such as sex, age, disease duration, autoantibodies (anti-CCP, RF), and prior DMARD use, which could have effects on the likelihood of patient response to treatment [34,69,75]. These data might be very relevant and affect our results. In the current study, eight studies reported that age did not influence MTX treatment response, and one study reported a lack of response, where age was consistent with what was reported earlier by Bologna et al., [76] who revealed that in RA, the age at initiation of MTX treatment did not influence its efficacy and toxicity. Similarly, eight studies reported that sex did not influence MTX treatment response, and one additional study reported that male patients were associated with an increased response to MTX. On the other hand, the disease duration effect was not reported by most of the included studies, except for seven studies, six of which reported that it did not influence MTX treatment response. One study reported that early disease indicated a better response than late disease, similar to data reported by Anderson et al., [75] who showed that RA patients with a longer disease duration do not respond well to MTX treatment compared to patients with early disease. Anti-CCP antibody and rheumatoid factor (RF) concentrations predicted greater disease activity in men with RA [77]. Furthermore, among patients with moderate disease activity (DAS28 > 3.2), anti-CCP but not RF was associated with joint damage or erosion [78] and the development of MTX resistance in early RA [79]. But low baseline disease activity was associated with a very low risk of radiographic progression [80]. The main difference between the clinical phenotypes was based on erosive disease [8]. However, with advanced therapy, these phenotypes showed less prominent differences in radiographic damage [71]. On the other hand, the long-term clinical outcomes and DMARD-free remission differ between the three RA clinical phenotypes. These data reconfirm that treatment might be stratified on these three phenotypes [81]. Unfortunately, the studies included in the current systematic review did not consider the three ERA clinical phenotypes as primary or secondary outcomes. Therefore, data on clinical phenotypes were not fully documented in our current study. Thus, only eight studies assessed the effect of clinical phenotype (anti-CCP/RF); seven of them reported that it did not influence MTX treatment response, and one study reported excellent response in RF-negative patients.

Additionally, smoking, which is a well-known environmental risk factor for RA [82], was shown to be associated with reduced response to MTX therapy [83,84]. Therefore, nor-nicotine was identified as a biomarker for inadequate response to MTX [6]. In the current study, only two studies reported data about smoking. One study reported that 20 RA patients were smokers [45] and stated that smoking did not influence MTX response, while the other study reported that only 4 patients were smokers [41], and that higher baseline levels of nor-nicotine were associated with no response to MTX, but statistical significance was not reached.

### 4.4. Our Recommendation for Methods

It is worth mentioning that in the current study, all the studies used only one biological fluid, which was blood. Three main methods were used to identify metabolomes:

Chromatography is a widely used technique for identifying and quantifying small-molecule metabolites. The only difference between liquid chromatography (LC) and gas chromatography (GC) is that the mobile phase in LC is liquid (slow and inexpensive), and in GC, there is an inert gas, such as helium (fast and expensive); the separated compounds (soluble and volatile, respectively) in both methods will be detected by mass spectrometry (MS). GC-MS-based metabolomics is ideal for identifying over 200 metabolites from human body fluids (e.g., plasma, urine, or stool) per study. The drawback is that the unit is very expensive, not widely available, and slightly complicated; thus, expert or trained personnel are required to use it. In the current study, GC-MS was used by one study [41]. LC-MS was used by one study [36], and MS-MS was used by one study [45]. High-performance liquid chromatography (HPLC) is an advanced form of LC and was used by two studies [46,49].

Immunoassays: Regarding the enzyme-linked immunosorbent assay (ELISA) and chemiluminescence immunoassay (CLIA), the ELISA, which is an easy and reliable method, was used by four studies [47,48,50,51]. The CLIA was used by only one study [38]. The differences between the two methods are that the ELISA measures optical density, is time-consuming, has low sensitivity, and is a less expensive test, while the CLIA measures relative light units, is rapid, has high sensitivity, and is an expensive test.

Fluorescence-activated cell sorting (FACS) and flow cytometry: FACS is a process by which a sample mixture of cells is sorted according to light scattering and fluorescence characteristics into two or more containers. Flow cytometry is a methodology that is utilized during the analysis of a heterogeneous population of cells according to different cell surface molecules, size, and volume, which allows for the investigation of individual cells. FACS, together with flow cytometry, can measure and characterize multiple cell generations by using highly specific antibodies tagged with fluorescent dyes. Using flow cytometry, we can determine cell phenotypes, cell functions, and even sort live cells. A researcher can perform FACS analysis and simultaneously gather expression data and sort cell samples by several variables. In the current study, FACS and flow cytometry were used by four studies [37,40,42,44].

Risk of bias (ROB) is a procedure that can improve the quality of systematic reviews and meta-analyses. This measure helps to report more accurate results. Our assessment of the three RCTs revealed that one study showed high ROB, and two studies showed unclear ROB. Out of the 13 cohort studies, 11 studies (84.6%) were judged to have an overall high ROB, while 2 studies (15.4%) were assessed as having some concerns regarding the overall ROB.

### 4.5. Strengths and Limitations of This Study

One of the strengths of the current study is that 12 out of the 16 studies are prospective (observational) cohort studies. There were three RCTs and only one retrospective cohort study. Another strength is that all the studies used changes in DAS28-ESR as a clinical assessment tool for MTX response. One more strength was that no prior DMARD was used, which could affect the likelihood of patient response to MTX treatment. Among the strengths is the importance of this study in clinical practice and the efforts undertaken to minimize the risk of bias. Moreover, we could recommend only 31 top predictive biomarkers among the 102; a few of them, which indicated remission, were recommended as first-line biomarkers for use in routine clinical practice after validation. We also provided the main methods to be used. One of the limitations of the current study was that the included studies were very heterogeneous in their methodology, and some studies lacked clarity in the method of detailed evaluation of their results. Most of the included studies did not compare their results between responders and non-responders with different RA clinical phenotypes and demographic data. Despite the considerably large analyzed cohorts and metabolites, the studies failed to identify a high characterization of RA responders, which was mainly due to the heterogeneity in the studies and the fact that each biomarker was tested by 1–2 studies, which made it difficult to compare the metabolites and make a general conclusion.

## 5. Conclusions

The current study revealed different types of biomarkers that can be recognized as potential indicators for MTX treatment response in early RA patients. Our study classified patients with RA in terms of expected therapeutic responsiveness into responders and non-responders; our results can guide drug therapy and act as a foundation for precision medicine to overcome the under- or overtreatment of early RA patients. Furthermore, we revealed that a few biomarkers (out of the 31) resulted in a state of remission. These biomarkers are itaconate and its derivatives, circulating monocytes and their three subset cells, serum IL-6 levels, serum levels of 11 endogenous metabolites, and functional multi-resistant protein (fMRP) together with reduced folate carrier (RFC). These biomarkers are promising but not yet ready for routine use. We recommend that clinicians focus on the biomarkers that achieved a remission state as first-line biomarkers for use in routine clinical practice. Further studies are still needed to identify subgroups or clinical RA phenotypes that will respond to MTX treatment or other treatment strategies. However, regarding the 31 biomarkers identified in our study, the implementation of personalized medicine in early RA patients still requires further validation in larger prospective trials. These biomarkers should be combined not only with baseline DAS28 scores but also with clinical phenotypes to identify which ones are applicable for daily clinical practice and for accurate prediction of MTX treatment response in early RA based on individual patient characteristics.

Future directions

We revealed that a few biomarkers resulted in a state of remission in patients with ERA. These biomarkers are promising but not yet ready for routine clinical use; they warrant validation in larger prospective trials. We recommend that for the implementation of personalized medicine, these biomarkers should be the first-line biomarkers for use in routine clinical practice after validation.

## Figures and Tables

**Figure 1 metabolites-15-00715-f001:**
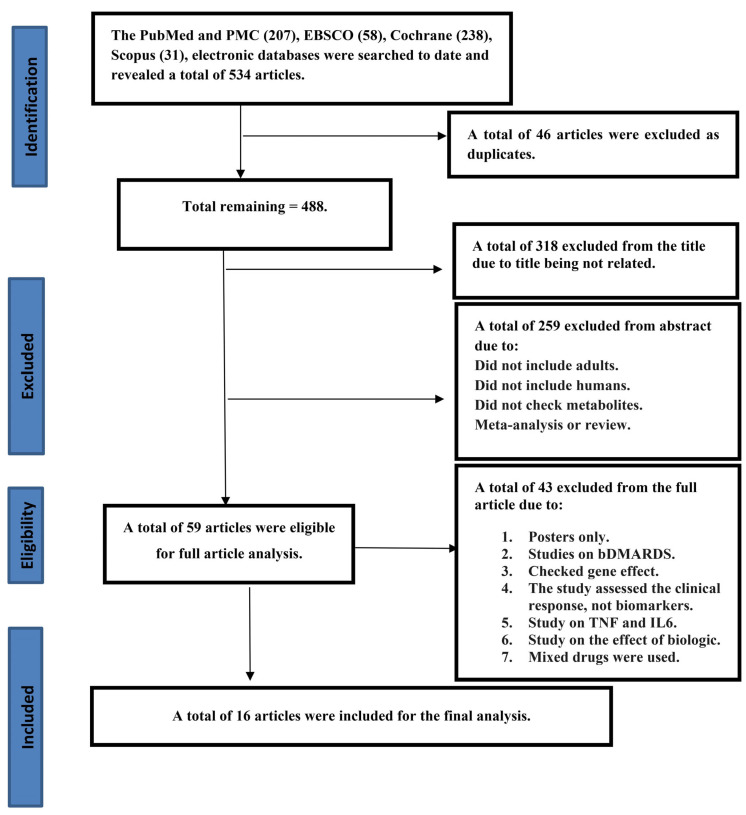
PRISMA 2020 diagram for predictive biomarkers of methotrexate treatment response in patients with rheumatoid arthritis: a systematic review.

**Figure 2 metabolites-15-00715-f002:**
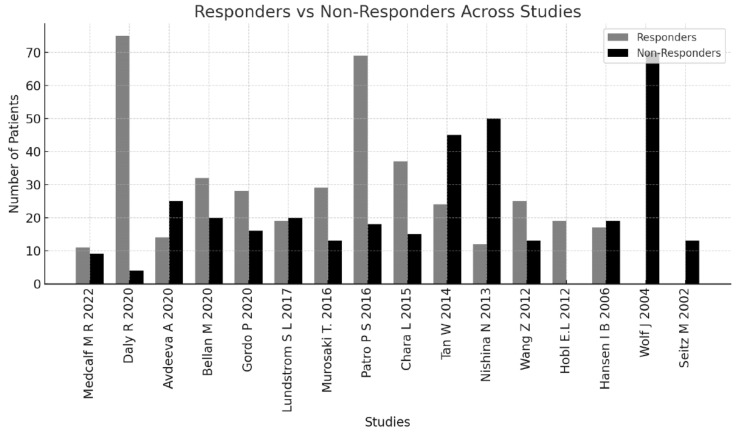
Responders vs. non-responders to methotrexate across studies. Citations in Chronological Order: Medcalf M R, et al., 2022 [41]; Daly R, et al., 2020 [36]; Avdeeva A, et al., 2020 [42]; Bel-lan M, et al., 2020 [43]; Gordo P F et al., 2020 [44]; Lundstrom S, et al., 2017 [45]; Murasaki T et al., 2016 [46]; Patro P S et al., 2016 [47]; Chara L, et al., 2015 [37]; Tan W, et al., 2014 [48]; Nishina N., et al., 2013 [38]; Wang, Z. et al., 2012 [39]; Hobl E L, et al., 2012 [49]; Hansen IB, et al., 2006 [50]; Wolf J, et al., 2004 [40]; Seitz M, et al., 2002 [51].

**Figure 3 metabolites-15-00715-f003:**
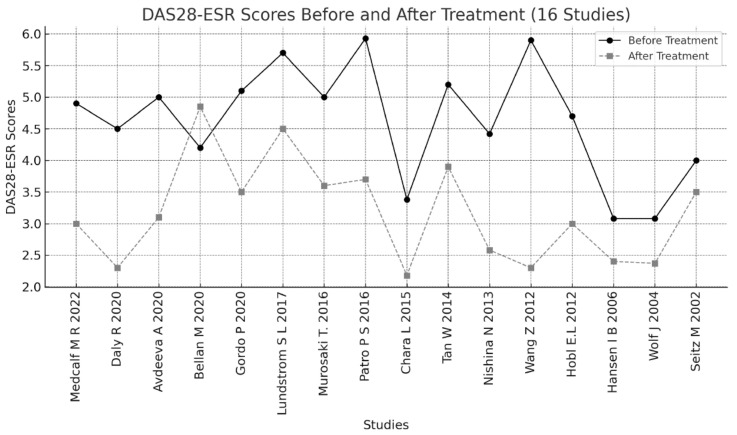
DAS28-ESR scores before and after methotrexate treatment. Citations in Chronological Order: Medcalf M R, et al., 2022 [41]; Daly R, et al., 2020 [36]; Avdeeva A, et al., 2020 [42]; Bel-lan M, et al., 2020 [43]; Gordo P F et al., 2020 [44]; Lundstrom S, et al., 2017 [45]; Murasaki T et al., 2016 [46]; Patro P S et al., 2016 [47]; Chara L, et al., 2015 [37]; Tan W, et al., 2014 [48]; Nishina N., et al., 2013 [38]; Wang, Z. et al., 2012 [39]; Hobl E L, et al., 2012 [49]; Hansen IB, et al., 2006 [50]; Wolf J, et al., 2004 [40]; Seitz M, et al., 2002 [51].

**Figure 4 metabolites-15-00715-f004:**
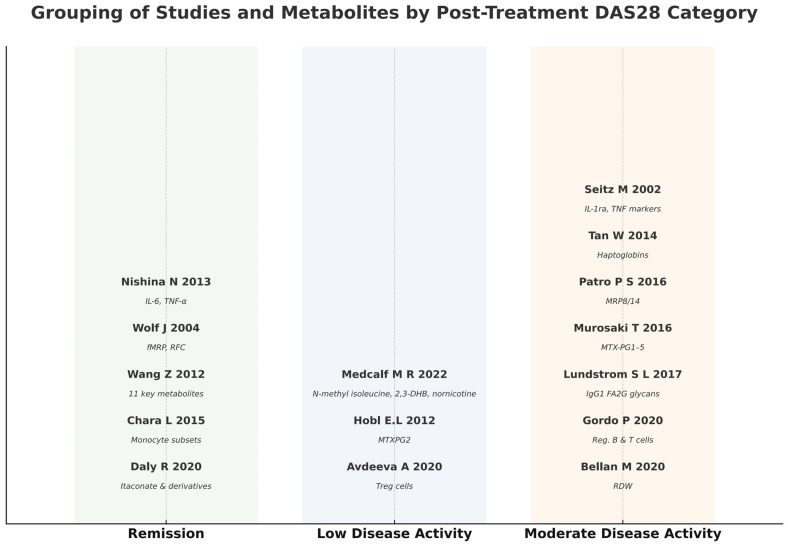
The grouping of the studies and their corresponding metabolites. Citations in Chronological Order: Medcalf M R, et al., 2022 [41]; Daly R, et al., 2020 [36]; Avdeeva A, et al., 2020 [42]; Bel-lan M, et al., 2020 [43]; Gordo P F et al., 2020 [44]; Lundstrom S, et al., 2017 [45]; Murasaki T et al., 2016 [46]; Patro P S et al., 2016 [47]; Chara L, et al., 2015 [37]; Tan W, et al., 2014 [48]; Nishina N., et al., 2013 [38]; Wang, Z. et al., 2012 [39]; Hobl E L, et al., 2012 [49]; Hansen IB, et al., 2006 [50]; Wolf J, et al., 2004 [40]; Seitz M, et al., 2002 [51].

**Figure 5 metabolites-15-00715-f005:**
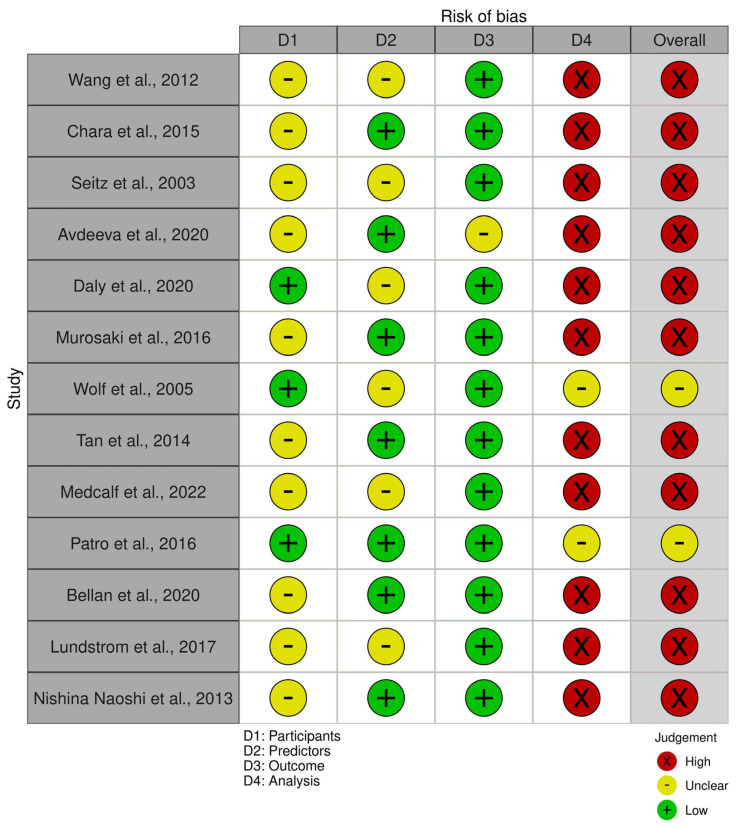
Risk of bias for observational studies.

**Figure 6 metabolites-15-00715-f006:**
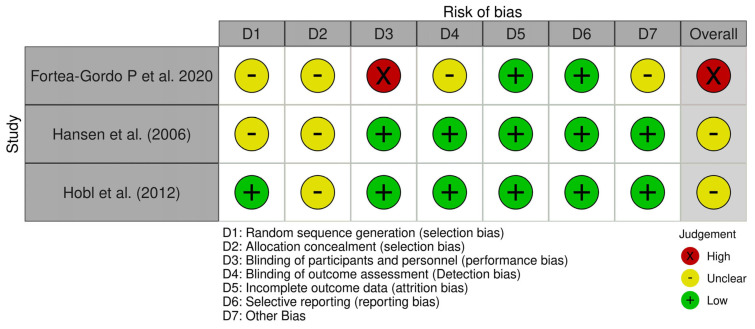
Risk of bias for randomized controlled trials.

**Table 1 metabolites-15-00715-t001:** Comparison of the current systematic review with existing systematic and scoping reviews on biomarkers in rheumatoid arthritis.

Aspect	Existing Systematic/Scoping Reviews	Current Systematic Review
Main Focus	Broadly evaluate metabolomics, clinical prediction models, or general biomarkers across various RA therapies (MTX, TNF inhibitors, IL-6 blockers, etc.).	Focused exclusively on predictive biomarkers of methotrexate (MTX) treatment response in early rheumatoid arthritis (ERA).
Objective	Summarize associations between biomarkers and disease activity or treatment response, often across multiple drugs.	Identify and validate metabolomic, proteomic, inflammatory, and immune cell biomarkers predictive of MTX response for precision medicine.
Study Type	Systematic or scoping reviews, some with meta-analyses (e.g., MTX–polyglutamates, clinical prediction models), heterogeneous in methods.	Systematic review, PROSPERO-registered (CRD42024547651), PRISMA-compliant with clearly defined eligibility and synthesis methods.
Data Scope	Typically, 10–25 studies including multiple RA therapies; focus on limited biomarker classes.	A total of 16 MTX-specific human studies (2000–2024); 946 patients; 102 biomarkers analyzed; 31 biomarkers identified as functionally predictive.
Therapeutic Scope	Include multiple therapeutic agents; MTX findings are secondary or indirect.	Dedicated entirely to MTX monotherapy in early RA.
Omic Integration	Usually single-omic focus (metabolomic or proteomic or genetic).	Integrates multi-omic domains: metabolomic, proteomic, inflammatory, and immune cell biomarkers.
Clinical Relevance	Rarely connect biomarkers with validated clinical indices (e.g., DAS28).	Directly links biomarker patterns to DAS28-based clinical outcomes and treatment response categories.
Novelty/Time Frame	Mostly include studies up to 2023; limited integration of recent findings, such as itaconate or IgG glycosylation.	Incorporates the latest evidence through 2024, including novel predictors (itaconate, FA2-IgG1, RFC/fMRP, MTX-PG1–7).
Outcome Orientation	Emphasize biological plausibility and mechanistic interpretation.	Provides a clinically actionable framework for individualized MTX therapy prediction.
Scientific Contribution	Offer general insights but lack MTX-specific predictive synthesis.	Advances the field from descriptive multi-drug biomarker reviews to an MTX-specific, multi-omic predictive precision medicine model.

**Table 2 metabolites-15-00715-t002:** Characteristics of rheumatoid arthritis patients and the articles included in this study.

No	1	2	3	4	5
	Author	Age in Years Mean Age ± SD Median Age (Range)	Total No of Patients (M:F)	Country	Type of Study/MC/SC
T	16 Studies	51.99 ± 5.79 (32.5–86)	946 (200:746)	11 Countries	RCT = 11 Pros = 4 Retros = 1
1	Medcalf M R, et al., 2022 [41]	53 (38–66)	20 (6:14)	USA	Pros/SC
2	Daly R, et al., 2020 [36]	56 ± 13	79 (25:54)	USA	Pros/SC
3	Avdeeva A, et al., 2020 [42]	52 (32.5–57.5)	45 (6:39)	Russia	Pros/SC
4	Bellan M, et al., 2020 [43]	62.5 (52–69)	82 (27:55)	Italy	Retros/SC
5	Gordo P F et al., 2020 [44]	54 (40.8–59.8)	48 (9:39)	Spain	RCT/SC
6	Lundstrom S, et al., 2017 [45]	53 (45–62)	59 (17:42)	Sweden	Pros/SC
7	Murasaki T et al., 2016 [46]	C1:66.0 (36.0–86.0)	37 (11:26)	Japan	Pos/SC
8	Patro P S et al., 2016 [47]	40 (33–50)	87 (13:74)	India	Pros/SC
9	Chara L, et al., 2015 [37]	Rs: 52.44 ± 10.90 NRs: 52.08 ± 10.87	52 (14:38)	Spain	Pros/SC
10	Tan W, et al., 2014 [48]	Rs 42.3 ± 9.4 NRs 43.6 ± 11.6	69 (5:64)	China	Pros/SC
11	Nishina N., et al., 2013 [38]	57 (NA)	62 (13:49)	Japan	Pros/SC
12	Wang, Z. et al., 2012 [39]	56.4 ± 2.8	38 (26:5)	China	Pros/SC
13	Hobl E L, et al., 2012 [49]	56 years	19 (6:13)	Austria	RCT/SC
14	Hansen IB, et al., 2006 [50]	55.1 years	36 (11:25)	Denmark	RCT/SC
15	Wolf J, et al., 2004 [40]	59.5 years	163 (47:116)	Austria	Pros/SC
16	Seitz M, et al., 2002 [51]	G1 = 50.9 ± 13.8 G2 = 52.2 ± 5.7 G3 = 55.9 ± 13.4 G4 = 52.9 ± 18.5	50 (G1 = 1:12) (G2 = 0:3) (G3 = 5:15) (G4 = 6:8)	Switzerland	Pros/SC

T = total; M:F = male–female; SD = standard deviation; MC/SC = multicenter/single center; RCT = randomized clinical trial; Pros = prospective observational cohort; Retros = retrospective cohort; NA = Not applicable; G1–4 = groups 1–4.

**Table 3 metabolites-15-00715-t003:** The patients’ different demographic and disease-associated variables.

No	1	2	3	4	5	6	7	8	9	10	11
	Author	No. of pts T = 946	Disease Duration Median (Range) M± SD	Study Duration	Treatment Duration 465 W M = 29.06 (Range 12–129)	Diagnostic Criteria of the pts	Types and Nos. of Biomarkers (Total No = 102)	Biomarkers Baseline in pts	Methods of Biomarker Measurement	Intervals at Measurement/Week	No of HC
1	Medcalf M R, et al., 2022 [41]	20	9 (4–14) Ms	NA	16 Ws	-ACR -DAS-28	19 plasma metabolomes	NA	(GC-MS)	-NA -16 Ws	NA
2	Daly R, et al., 2020 [36]	79	5.3 ± 3.1 Ms	3 Ms	12.9 Ws (3 Ms)	-2010 ACR/EULAR DAS44-ESR	9 plasma metabolites	NA	LC-MS Westergren method (ESR), RIA, ELISA, Nephelometry (CRP)	-0 W -12.9 Ws	NA
3	Avdeeva A, et al., 2020 [42]	45	5 (4–6) Ms	24 W	24 Ws	2010 ACR/EULAR	1 (CD4^+^ FoxP3^+^ Treg cells)	NA	FACS analysis ELISA and multiplex immunoassays	-12 Ws -24 Ws	20
4	Bellan M, et al., 2020 [43]	82	6 (2–26) Ms	January 2010–December 2018	24 Ws 3 Ms	2010 ACR/EULAR	1 (RDW)	13.9% (13.1–14.8)	XN 2000 hematology analyzer sysmex	-0 Ws -12.9 Ws	NA
5	Gordo P, et al., 2020 [44]	48	4 (2–6) Ms	NA	51.6 Ws 12 Ms	2010 ACR Revised	5 (3 B cells, 2 T cells)	NA	Flow cytometry	-0 Ws -51.6 Ws	48
6	Lundstrom S L, et al., 2017 [45]	59	<1 yr	1996–2006	14 Ws (13–15)	EULAR	19 (Fc glycopeptides, IgG1-4)		MS-MS	-0 Ws -14 Ws	11
7	Murosaki T. et al., 2016 [46]	42 (C1)	C1: 0.3 (0.0–24.5) yrs	July 2013–November 2015	24 Ws	EULAR DAS28-CRP	5 (MTXPG 1-5)	NA	HPLC	-0 Ws -8 Ws -24 Ws	NA
8	Patro P S et al., 2016 [47]	87	28 Ms	February 2014–May 2015	17.2 Ws 4 Ms	EULAR 2010	1 (MRP8/14)	19.95 µg/mL (11.49–39.06)	ELISA	-0 Ws -8 Ws -16 Ws	
9	Chara L, et al., 2015 [37]	52	<12 Ms	6 Ms	25.8 Ws 6 Ms	EULAR DAS 28	4-PBMC and its 3 subsets (CD14^+high^^/low^ CD16^+/−^)	Low no. of PBMC in Rs	Flow cytometry	-0 Ws -12 Ws -24 Ws	15
10	Tan W, et al., 2014 [48]	69	<24 Ms	1 yr	12 Ws	ACR 1987	1 (Haptoglobin, Hap)	Hap mg/dL in: Rs (255.3 ↓ to 143.9) NRs (369.9 ↓ to 159.8)	-ABI TaqMan (mRNA) -ELIZA (Hap)	-0 Ws -12 Ws	36
11	Nishina N., et al., 2013 [38]	62	<36 Ms	18 Ms	21.5 Ws 1 year	DAS 28	2 (IL-6 and TNF-α)	IL-6 (4.72 ↓ to 1.04, *p* < 0.001) TNF (0.87 ↓ to 0.83, *p* < 0.14)	Chemiluminescent enzyme immunoassay	-0 Ws -1 yr	-
12	Wang Z., et al., 2012 [39]	38	4.5 ± 1.9 yrs	24 Ws	24 Ws	ACR	20 (11 significant ** and 9 non *)	-0 Ws -11 Ws (*p* < 0.05) -24 Ws (*p* < 0.01)	^1^H NMR	-0 Ws -11 Ws -24 Ws	20
13	Hobl E.-L, et al., 2012 [49]	19	-	2008–2009	17 Ws 16 Ws	ACR	7 (MTXPG1-7)	-MTXPG2 Cmax 11.06 ± 12.51 ↑ to 22.06 ± 5.61	HPLC	-0 Ws -5 Ws -10 Ws -16 Ws	-
14	Hansen I B, et al., 2006 [50]	36	13.3 ± SD years	-	28 Ws	ACR	1 (p-CXCL12)	p-CXCL12 1855 ± 145 (*p* < 0.001)	ELISA	-0 Ws -16 Ws -28 Ws	50
15	Wolf J, et al., 2004 [40]	163	7.6 ± SD years	-	129 Ws 30 Ms	ACR	2 (fMRP, RFC)	NA	-RT-PCR -Flow cytometry	-0 Ws -12 Ws	-
16	Seitz M, et al., 2002 [51]	50	53.5 ± 7.12 year	-	24 Ws	ACR	5 (IL-1ra, IL1B, TNF-α sTENFR)	-G1–G4 = IL-1ra/IL-1ß > 100 -G1/G2 ↓ < 100	ELISA	-0 Ws -12 Ws -24 Ws	-

pts = patients; Ws = weeks; Ms = months; year = yr; NA = Not applicable; NRs = non-responders; Rs = responders; MTX = methotrexate; C1 = category1 of MTX naïve; DAS28 = disease activity score of 28 joints; ACR = American College of Rheumatology; EULAR = European League Against Rheumatism; ABI TaqMan = Applied Biosystem TaqMan; MTX PGs = MTX–polyglutamate concentration in erythrocytes; HPLC = high-performance liquid chromatography; HCs = healthy controls; IL-1ra1 = interleukin 1 receptor antagonist; sTNFR p55 + p75 = soluble tumor necrosis factor receptor p55 + 75; ^1^H NMR = nuclear magnetic resonance; MS = mass spectrometry; PBMCs = peripheral blood mononuclear cells; CCR7 = CC chemokine receptor 7; CCL19 = CC chemokine ligand 19; Mfi = mean fluorescence intensity; MMP-3 = matrix metalloproteinase-3; significant (11) ** = α-oxoglutarate, aspartate, methionine, histidine, hypoxanthine, glycine, taurine, tryptophan, trimethylamine-N-oxide (TMAO), uracil, uric acid; non-significant (9) *: citrate, acetate, alanin, cholesterol, creatinine, cysteine, lactate, glutamate, serine; FACS = fluorescence-activated cell sorting.

**Table 4 metabolites-15-00715-t004:** Responders and non-responders of rheumatoid arthritis patients to methotrexate therapy.

N	1	2	3	4	4	6	7	8
	Author	ACR/EULAR Response Criteria DAS28-ESR Before MTX (Baseline) M ± SD Median (Range)	ACR/EULAR Response Criteria DAS28-ESR After MTX M ± SD Median (Range)	Responders (Rs) No. of pts	Non-Responders (NRs) No. of pts	Positive ACCP or RF	MTX Rx Fixed Oral Dose/W	Other Therapy
1	Medcalf M R, et al., 2022 [41]	DAS28-ESR 4.9 (4.1, 6.0) -PGA	ΔDAS28-ESR: 3.0 (1.7–4.7) Rs: ∆ − 3.6 (−4.6−2.7)	11 pts	9 pts	NA	20 mg Oral/ SC	Folic acid
2	Daly R, et al., 2020 [36]	-DAS44-ESR -4.5 ± 1.2 -HAQ, ESR, CRP	DAS44 ESR 2.3 ± 1.3	75 pts	4 pts	-53% CCP^+^ -65% RF	-	Steroid SSZ
3	Avdeeva A, et al., 2020 [42]	DAS28 median 5.0 (4.2–5.8) -SDAI	DAS28 3.1 (2.7–3.62)	14 pts	25 pts	40 CCP^+^ 34 RF^+^	25 mg	NA
4	Bellan M, et al., 2020 [43]	DAS28-ESR 4.20 (3.27–5.03)	DAS28-ESR good Rs 4.85 (4.19–5.55)	-32 good Rs -20 moderate Rs	30 pts	-	15 mg	Steroid HCQ
5	Gordo P, et al., 2020 [44]	DAS28-ESR median: 5.1 (IQR 4.3–5.9)	NA	28 pts (64%)	16 pts	41 pts ACCP^+^/RF^+^	25 mg	Steroid HCQ
6	Lundstrom S L, et al., 2017 [45]	EULAR DAS28-ESR -median 5.7 (5.0–6.2) -HAQ	DAS28-ESR 4.5 (2.6–5.1)	-19 good Rs -20 moderate Rs	20 pts	44 pts (75%) CCP^+^/RF^+^	20 mg	NSAIDs Steroid
7	Murosaki T. et al., 2016 [46]	EULAR DAS28-CRP C1 5.0 (3.2–8.1)	C1 NA ∆ DAS28 > 1.2	29 pts	13 pts	CCP^+^C1 RF^+^C1 = 51.7%	16 mg	DMARDs Steroid NSAIDs Folic acid
8	Patro P S, et al., 2016 [47]	EULAR 2010 DAS28-ESR 5.93 (5.4–6.5) -Rs: 5.95 (5.3–6.5) -NRs: 5.92 (5.6–6.6) -HAQ	DAS28 2.86 (2.4–3.7)	69 pts	18 pts	-69 (79) ACCP^+^ -71 (82) RF^+^	25 mg	Folic acid NSAID
9	Chara L, et al., 2015 [37]	EULAR-DAS28 -Rs: 3.38 ± 0.54 -NRs: 3.55 ± 0.76 -HAQ, CRP	EULAR-DAS28 Rs = 2.18 ± 0.44 NRs 3.49 ± 0.20	37 pts	15 pts	-CCP^+^ -RF^+^	20 mg	Folic acid NSAIDs
10	Tan W, et al., 2014 [48]	ACR-DAS-28 -Rs: 5.2 4.8 ± 0.7 -NRs: 4.8 ± 0.9 -TSS, HAQ	ACR criteria DAS-28 -Rs: 3.9 ± 1.1 -NRs: 4.6 ± 1.2 -TSS, HAQ	24 pts	45 pt	36 pts (52.3%) ACCP	20 mg	NSAID HCQ
11	Nishina N, et al., 2013 [38]	DAS28 4.42 (3.60–5.62) SS, HAQ, MMP-3	DAS28 2.58 (1.93–3.16) -TSS, HAQ, MMP-3	12 pts	50 pts	-43 pts RF^+^/CCP^+^ -3 pts RF/CCP	Median: 8 mg	DMARDs SSZ TAC BUC Steroid
12	Wang Z, et al., 2012 [39]	ACR-DAS28 5.9 ± 1.4	ACR DAS28 Rs: 2.3 ± 0.8 NRs: 4.6 ± 1.2	25 pts	13 pts	38 CCP^+^ /RF^+^	10 mg	No other therapy
13	Hobl E L, et al., 2012 [49]	ACR-DAS28 -4.7 ± NA -HAQ (1.5)	ACR criteria 3.0 ± NA reduced	19 pts	None	42% FR^+^	25 mg	Steroid Folic acid
14	Hansen I B, et al., 2006 [50]	ACR response criteria	Reduced ACR Criteria independent of metabolite	17 pts	19 pts		7.5 mg	Steroid NSAIDs
15	Wolf J, et al., 2004 [40]	EULAR-DAS28 -G1 = 3.08 ± 1.02 -G2 = 3.22 ± 1.30 -G3 = 3.38 ± 1.10 -G4 = 3.39 ± 0.95	EULAR-DAS28 -G1 = 2.37± 1.0 -G2 = 1.99 ± 0.93 -G3 = 2.05 ± 1.14 -G4 = 2.30 ± 0.98	-32 (G1) good Rs -33 (G2) good Rs -28 (G3) moderate Rs	70 pts (G4)	-	12.2 mg	Steroid HCQ SSZ, CsA NSAIDS Folic acid
16	Seitz M, et al., 2002 [51]	ACR response criteria	ACR response criteria	14 (G1*) excellent 20 (G2*) good Rs	13 (G4*) NRs 3 (G3*) poor Rs	-	15 mg	Steroid NSAIDs

M = mean; SD = standard deviation; NA = not applicable; DA = disease activity; pts = patients; W = weeks; Ms = months; Sc = subcutaneous; ACCP = anti-cyclic citrullinated peptide; CsA = Ciclosporin/Cyclosporine A; HCQ = hydroxychloroquine; SSZ = Sulphasalazine; TAC = Tacrolimus; BUC = Bucillamine; HAQ = Health Assessment Questionnaire; DAS28 = disease activity score of 28 joints; MTX Rx = treatment; PGA = Patient Global Assessment; EULAR = European League Against Rheumatism; ACR = American College of Rheumatology; ACR response criteria = excellent ACR > 70; good ACR 50–70; poor ACR 20–50; non-responders ACR < 20; Rs = responders; NRs = non-responders; TSS = van der Heijde-modified Total Sharp Score; G1–G4 = groups 1–4; C1 = Category 1 (MTX-naïve pts).

**Table 5 metabolites-15-00715-t005:** Numbers, types, and outcomes of the predictive metabolomes of MTX treatment response in patients with early rheumatoid arthritis.

No	Author	Numbers and Types of Metabolomes Studied (Total No = 102)	Final No. 31	Outcomes
1	Medcalf M R, et al., 2022 [41]	19 plasma metabolites (nornicotine, N-methylisoleucine, and 2,3-dihydroxybutanoic acid, etc.)	3	Lower pre-treatment plasma levels of only three metabolites were associated with a greater reduction in DAS28-ESR (N-methyl isoleucine (0.54, *p* = 0.02), 2,3-dihydroxy butanoic acid (0.51, *p* = 0.02), and nor-nicotine (0.50, *p* = 0.02).
2	Daly R. et al., 2020 [36]	9 plasma metabolites: itaconate and its derivatives (itaconate anhydride, CoA), several peptides, cholesterol, and fatty acids	1	Increased levels of itaconate and its derivatives in response to MTX treatment demonstrated a consistent reduction in disease activity level (measured by DAS44-ESR and CRP).
3	Avdeeva A. et al., 2020 [42]	1 (CD4^+^ FoxP3^+^ Treg cells)	1	The defect in regulatory T cell (Treg) compartment negatively correlated with both RA activity and antibody level. MTX Rx of pts with early RA increased both the proportion and absolute number. Treg was a specific cellular marker of successful RA Rx.
4	Bellan M. et al., 2020 [43]	1 (RDW)	1	A larger baseline RDW was associated with poorer treatment response at 12 weeks, but the larger the increase in RDW from baseline after MTX initiation, the better the pt’s response to it.
5	Fortea-Gordo P et al., 2020 [44]	5 -3 B cells (cTrB, cMatN B cells, cMem B cells). -2 T cells (CD4^+^CDSRA^−^CD25^−^CXCR5^+^ and CD4^+^CD4SRA^−^CD25^−^).	1	Higher baseline frequencies of circulating regulatory B (cTrB) cells, but not cMatN or cMem B cells, were associated with a good EULAR response to MTX.
6	Lundstrom S L, et al., 2017 [45]	19 (Fc IgG1 (12), Fc IgG2 (7))	1	A baseline low level of galactosylated glycans (FA2G) of IgG1 in ERA (mainly low ratio of FA2/(FA2G1 + FA2G2) of IgG1) was significantly associated with nonresponse.
7	Murasaki T et al., 2016 [46]	5 (MTX-PG1-5)	1	MTX-PGs (PG1-5) in erythrocytes were potential indicators and predictors of MTX efficacy. The routine measurement of MTX-PGs by HPLC was difficult to perform in clinical practice.
8	Patro P S et al., 2016 [47]	1 (MRP8/14)	1	Serum MRP8/14 levels were correlated with disease activity at baseline and reduced on treatment with MTX in pts who responded to treatment. Thus, higher baseline MRP8/14 levels were associated with good response to MTX treatment.
9	Chara L, et al., 2015 [37]	4: PBMCs and 3 subsets of (CD14^+high^CD16^−^ CD14^+high^ CD16^+^ CD14^+low^CD16^+^)	3	In untreated pts, a higher pre-treatment number of circulating monocytes and higher numbers of cell subsets (CD14^+high^CD16^−^, CD14^+high^CD16^+^ and CD14^+low^CD16^+^) provide good predictive biomarkers of a reduced clinical response to MTX.
10	Tan W, et al., 2014 [48]	1 (3 Haptoglobins, Hap)	1	High serum levels of Hap at baseline were associated with the inadequate response of 12-week MTX treatment in early RA pts but could not predict the structural damage at one year.
11	Nishina N, et al., 2013 [38]	2 (IL-6, TNF-α pg/ml)	1	Serum IL-6 levels significantly reduced after MTX treatment in early RA pts, while TNF-α plasma level did not change. A high plasma concentration of IL-6 after MTX was the parameter that was most associated with radiographic progression.
12	Wang, Z. et al., 2012 [39]	20 (11 significant ** and 9 non-significant *)	11	Serum levels of 11 endogenous metabolites of the effective group showed a significant difference when compared with those of the non-effective group (*p* < 0.05) and correlated with good MTX response in pts with early RA.
13	Hobl E.-L, et al., 2012 [49]	7 (MTXPG 1-7)	1	High pre-treatment level of short-chain MTXPG2 was revealed to be a potential biomarker for good clinical outcome in RA pts, and C_max_ positively correlated with the improvement in DAS-28 (R = +0.518, *p* = 0.023).
14	Hansen I B, et al., 2006 [50]	1 (p-CXCL12)	1	The p-CXCL12 level was constantly high and independent of any ACR disease activity variables, as well as response to MTX treatment. Indicated no response.
15	Wolf J, et al., 2004 [40]	2 (fMRP and RFC)	2	The lack or the presence of both fMRP and RFC (fMRP+/RFC+ and fMRP−/RFC−) led to a significantly better therapeutic outcome.
16	Seitz M, et al., 2002 [51]	5 (IL-1ra, IL-1B, sTENFRp55, sTENFRp75, TNF-a)	1	Constitutively increased IL-1ß produced by PBM significantly lowered the ratio of IL-1ra/IL-1ß (<100, *p* < 0.00001), which was associated with good and excellent responses to MTX.

RA = rheumatoid arthritis; pts = patients; MTX = methotrexate; Treg cells = regulatory T cells; TrB cell = transitional regulatory B cell CD19^+^CD27^−^CD24^high^CD38^high^; Mem B cell = memory B cells (CD19^+^CD27^−^CD24^high^CD38); MatN B = mature naïve B cells (CD19^+^CD27^−^CD24^low^CD38^low^); Hap = haptoglobin; DAS = disease activity score; ESR = erythrocyte sedimentation rate; MTX PGs = methotrexate–polyglutamate concentration in erythrocytes; RDW = red cell distribution width; IL-1ra1 = interleukin 1 receptor antagonist; sTNFR p55 + p75 = soluble tumor necrosis factor receptor p55 + 75; PBMCs = peripheral blood mononuclear cells (CD14^+high/low^CD16^+/−^,); CCR7 = CC chemokine receptor 7; CCL19 = CC chemokine ligand 19; MMP-3 = matrix metalloproteinase-3; fMRP = functional multi-resistant protein; RFC = reduced folate carrier; 11 significant ** = α-oxoglutarate, aspartate, methionine, histidine, hypoxanthine, glycine, taurine, tryptophan, trimethylamine-N-oxide (TMAO), uracil, uric acid; 9 non-significant * = citrate, acetate, alanine, cholesterol, creatinine, cysteine, lactate, glutamate, serine; p-CXCL12 = plasma chemokine CXCL12; MRP8/14 = myeloid-related proteins; C_max_ = maximum plasma concentration.

**Table 6 metabolites-15-00715-t006:** Categorized summary of predictive biomarkers for methotrexate treatment response in early rheumatoid arthritis.

Category	Up Regulated in Good Responders	Downregulated in Good Responders	Interpretation/Functional Notes
Metabolites and Amino Acids	Choline, inosine, hypoxanthine, guanosine, nicotinamide, diglyceride	N-methyl-isoleucine, 2,3-dihydroxy-butanoic acid, nor-nicotine, glucosyl-ceramide, itaconic acid	Higher energy-related metabolites and purines indicate effective MTX metabolism and anti-inflammatory adenosine signaling; decreased branched-chain and lipid-derived acids associate with favorable response.
Fatty Acid/Lipid Pathway Metabolites	Diglycerides, nicotinamide-linked phospholipids	Glucosylceramide, itaconate pathway intermediates (itaconate, itaconate anhydrase, itaconate CoA)	Altered lipid turnover reflects anti-inflammatory lipid remodeling; accumulation of itaconate derivatives predicts poor MTX response.
Serum Proteins/Enzymes	MRP8/14 complex (myeloid-related protein 8/14), short-chain MTX–polyglutamates (MTX-PG1–7, esp. PG2), functional multi-resistant protein (fMRP) + reduced folate carrier (RFC) co-expression	Haptoglobin (Hap)	Elevated baseline inflammatory MRP8/14 and efficient intracellular MTX transport (fMRP + RFC, MTX-PGs) predict good response; high acute-phase haptoglobin indicates non-response.
Immunoglobulin and Glycoprotein Markers	—	FA2 glycoform of IgG1 (low FA2/(FA2G1 + FA2G2) ratio → non-response)	Aberrant IgG1 Fc glycosylation linked to persistent inflammation and poor therapeutic outcome.
Immune Cell/Hematologic Parameters	Higher baseline circulating transitional regulatory B cells (cTrB); lower baseline Treg counts normalize post-therapy; moderate rise in red cell distribution width (RDW) after MTX initiation	High baseline monocyte counts (CD14^+^high/low CD16^−^/^+^ subsets)	cTrB enrichment and Treg restoration signify effective immune regulation; excess monocytes correlate with inadequate response.
Cytokine/Inflammatory Ratios	Lower IL-1ra/IL-1β ratio after MTX therapy	High IL-6, CRP, CCL19 before treatment → risk of poor outcome	Post-treatment IL-1β dominance indicates successful cytokine suppression cascade and therapeutic efficacy.

## Data Availability

The original contributions presented in this study are included in this article; further inquiries can be directed to the corresponding authors.

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
