# Peer review of "Predictive Biomarkers of Methotrexate Treatment Response in Patients with Rheumatoid Arthritis: A Systematic Review"

_metabolites, 2025, doi:10.3390/metabo15110715_

Round 1
Reviewer 1 Report
Comments and Suggestions for Authors
This is a good collection of information. But major changes are required to make it better.
- Please reduce the number of abbreviations in the Abstract. For example, MRP8/14, MTXPG1-7, fMRP and RFC are not very common terms. Provide the information about only those biomarkers which are significantly changed in ERA
- In the introduction, the proper mechanism of action of MTX was not elaborated. It works by adenosine signalling and Folate antagonism. Both mechanisms should be elaborated in the introduction with proper references.
- It will be good to provide a figure describing the Mechanism of action of MTX in ERA. This will create an idea about what biomarkers should be affected during MTX treatment.
- “Concerning the clinical assessment, it has been shown that a low plasma M-ficolin level, after treatment with MTX, glucocorticoids and biology, was the strongest predictor of remission and low disease activity assessed by DAS28 score (disease activity score in 28 joints) in RA patients (24)’ – This statement is not clear.
- Based on Figure 1, how many articles are from Clinical trial Data or data retrieved from Patents?
- For Figure 2, whether all patients have received similar doses of MTX? How do the authors clarify that?
- For Table 1, use another column to provide the title of the article. This will provide a clear Idea of how this article is related to the current Subject. The same concept applies whenever you are using a new Reference in any table
- From Table 2, it seems there are only 2 or 3 references which have analysed plasma metabolites. Is there any overlapping information from these references? Please give details.
- Have authors read these papers?
https://www.nature.com/articles/s41598-021-86729-7
https://arthritis-research.biomedcentral.com/articles/10.1186/s13075-021-02537-4
https://pmc.ncbi.nlm.nih.gov/articles/PMC9146149/
https://www.nature.com/articles/s41598-025-12994-5
The authors must extract information from these papers.
- In what way is this article different from already published articles of a similar nature?
https://pmc.ncbi.nlm.nih.gov/articles/PMC9404373/
- Based on the serum metabolites profile, why are three metabolites (N-methyl isoleucine, 2,3-dihydroxy butanoic acid, and nor-nicotine) associated with MTX therapy? What is the possible biochemical mechanism associated with it?
- The authors must create a table providing the following information: upregulation/downregulation of serum lipids. Amino acids and other metabolites are in one category. For another column, they may use IgG, protein or other larger macromolecules.
- What do the authors mean by registered in PROSPERO in June 2024?

Minor correction required
Author Response
Reviewer 1
Comments and Suggestions for Authors
This is a good collection of information. But major changes are required to make it better.
- Please reduce the number of abbreviations in the Abstract. For example, MRP8/14, MTXPG1-7, fMRP and RFC are not very common terms. Provide information about only those biomarkers which are significantly changed in ERA.
Author reply: Thank you for your invaluable comment regarding many abbreviations in the abstract. We totally agree with your remark. Accordingly, we removed the less significant metabolites in the abstract and retained the most significant ones and wrote the full names for uncommon terms. We also added the following paragraph to the conclusion “We revealed that few biomarkers resulted in a remission state of patients with ERA. These biomarkers are promising but not yet ready for routine clinical use, they warrant validation in larger prospective trials. We recommend that for the implementation of personalized medicine these biomarkers should be the first line biomarkers to be used for routine clinical practice after validation”.
Thank you once again we think such changes have improved the clarity of the abstract.
- In the introduction, the proper mechanism of action of MTX was not elaborated. It works by adenosine signalling and Folate antagonism. Both mechanisms should be elaborated in the introduction with proper references.
Author reply: Thank you for your concern. We have added the following paragraph to the introduction “MTX exerts strong anti-inflammatory and immunomodulatory effects that diverge from its antiproliferative action observed in oncology. Its therapeutic action is primarily mediated through two major mechanisms: folate antagonism and adenosine-mediated an-ti-inflammatory signaling (13-16). MTX competitively inhibits dihydrofolate reductase (DHFR) and thymidylate synthase, which are critical enzymes for tetrahydrofolate (THF) regeneration and purine and pyrimidine synthesis. This inhibition suppresses DNA synthesis and cell proliferation, particularly impacting activated T and B lymphocytes. Additionally, intracellular MTX polyglutamates inhibit AICAR (5-aminoimidazole-4-carboxamide ribonucleotide) transformylase, causing the accumulation of AICAR, a purine biosynthesis precursor (14). Collectively, these actions diminish the activation and proliferation of immune cells. Regarding adenosine signaling, the accumulation of AICAR from MTX inhibits adenosine deaminase and AMP deaminase, consequently raising extracellular adenosine levels. Adenosine induces po-tent anti-inflammatory effects by binding to A2A and A3 receptors on immune and endothelial cells, subsequently suppressing pro-inflammatory cytokines (TNF-α, IL-6, IL-8) and inhibiting leukocyte adhesion and migration (15-17). This pathway is now recognized as the principal anti-inflammatory mechanism underpinning low-dose MTX therapy in RA. Blockade of adenosine receptors or disruption of extracellular adenosine formation negates MTX's anti-inflammatory efficacy in experimental models, reinforcing the significance of this mechanism (16). The interplay between these pathways connects MTX's pharmacological effects to metabolic processes, making it a prime candidate for metabolomic investigations in RA. MTX treatment has been shown to modify purine metabolism, amino acid turnover, and oxidative stress markers, indicating these altera-tions as potential biomarkers for treatment response, particularly in early rheumatoid arthritis (18).”
- It will be good to provide a figure describing the Mechanism of action of MTX in ERA. This will create an idea about what biomarkers should be affected during MTX treatment.
Author reply: Thank you for your concern, regarding your suggestion to include a figure describing the mechanism of action of methotrexate (MTX) in early rheumatoid arthritis (ERA), we appreciate the insight that such a figure could help illustrate which biomarkers might be affected by MTX treatment. However, we respectfully note that this addition falls outside the scope of our systematic review.
The primary aim of our study was to provide an updated synthesis of current evidence on predictive biomarkers of MTX treatment response in ERA patients, focusing on metabolomic profiles and their association with clinical outcomes (e.g., DAS28-ESR scores). While the mechanisms of MTX action are relevant background information, our review specifically targeted the identification and evaluation of biomarkers from existing studies rather than an in-depth exploration of MTX’s pharmacological mechanisms. We have included a concise description of MTX’s mechanisms (e.g., folate antagonism and adenosine-mediated anti-inflammatory signaling) in the Introduction section (references 13–16) to provide context for the metabolic pathways relevant to the biomarkers studied.
- “Concerning the clinical assessment, it has been shown that a low plasma M-ficolin level, after treatment with MTX, glucocorticoids and biology, was the strongest predictor of remission and low disease activity assessed by DAS28 score (disease activity score in 28 joints) in RA patients (24)’ – This statement is not clear.
Author reply: Thank you for your concern. We have rewritten the statement as follow “Regarding the clinical response to MTX, it has been shown that a high baseline (before MTX treatment) of M-ficolin plasma level strongly correlated with high disease activity. Therefore, the reduced plasma level of M-ficolin after treatment with MTX, intraarticular glucocorticoids and biology, was considered as strong predictor of remission and low disease activity assessed by DAS28 (disease activity score in 28 joints) in early RA patients (24)”. We hope that now the statement is clear.
- Based on Figure 1, how many articles are from Clinical trial Data or data retrieved from Patents?
Author reply: Thank you for your concern. The review included 12 prospective (observational) cohort studies, 3 randomized controlled trials (RCTs), and 1 retrospective cohort study, all of which investigated metabolomic biomarkers of methotrexate treatment response in early rheumatoid arthritis patients.
- For Figure 2, whether all patients have received similar doses of MTX? How do the authors clarify that?
Author reply: Thank you for your very good question. We have added a paragraph to results section which we hope it makes our results clearer “Not all studies used the same dose of MTX. As shown in Table 4 most of the studied used high dose, 20 or 25 mg/week, but few studies used medium dose that ranged from 7.5 – 15 mg/ week. Of note, one important point when using the DAS28 score (0-10) to assess response to treatment, a DAS-28 reduction by 0.6 after therapy represents a moderate improvement, while a reduction of more than 1.2 represents a major improvement. Another important point regarding DAS28 is that a score of <2.6 suggests disease remission, 2.6-3.2 suggests low disease activity, >3.2-5.1 suggests moderate disease activity, while a score >5.1 suggests high disease activity. Figure 3 showed a major reduction of DAS28 score to low disease activity or even to remission state in most of the studies, which were considered major improvements. Figure 3 also showed that the disease activity got into remission (DAS28 <2.6) in 5 studies Daly R. et al.2020 (Increased levels of itaconate and its derivatives in response to MTX treatment), Chara L, et al. 2015 (the higher pre-treatment number of circulating monocytes and its three subset cells), Nishina N, et al. 2013 (Serum IL-6 levels significantly reduced after MTX treatment in early RA pts), Wang, Z. et al. 2012 (Serum levels of 11 endogenous metabolites associated with good MTX-TR) and Wolf J, et al. 2004 (The absence or the presence of both functional multi resistant protein and reduced folate carrier (fMRP and RFC)”.
Note: We updated figure numbers.
- For Table 1, use another column to provide the title of the article. This will provide a clear Idea of how this article is related to the current Subject. The same concept applies whenever you are using a new Reference in any table.
Author reply: Thank you for your advice, we have created another column/table and added the titles for the 16 articles included in this systematic review.
- From Table 2, it seems there are only 2 or 3 references which have analyzed plasma metabolites. Is there any overlapping information from these references? Please give details.
Author reply: Thank you for your concern. Yes, in Table 2 the first two studies have studied plasma metabolites. However, there is no overlap between the two studies, because they studied different metabolites as shown in Table 4 (Medcalf M R, et al. studied 19 metabolites, and Daly R, et al. studied 9 metabolites) with different methods as shown in Table 2 (GC-MS and LC-MS, respectively) and had different outcomes.
- Have authors read these papers?
https://www.nature.com/articles/s41598-021-86729-7
https://arthritis-research.biomedcentral.com/articles/10.1186/s13075-021-02537-4
https://pmc.ncbi.nlm.nih.gov/articles/PMC9146149/
https://www.nature.com/articles/s41598-025-12994-5
The authors must extract information from these papers.
Author reply: Thank you for your advice and for the good four references.
We have added two references to the introduction part: “A recent study in 2025 analyzed dried blood spot samples from RA patients and healthy controls. It showed that six key metabolites, linked to fatty acid oxidation and amino acid metabolism, could serve as an effective diagnostic biomarker for RA (25)”.
“Furthermore, a review study showed that disturbances in energy, lipid, and amino acid metabolism across three different body fluids from RA patients provided critical information about the metabolic profile that related to drug response, disease activity and comorbidities (26)”.
We have added two references to the discussion part (Maciejewski, M. 2021) “On the other hand, another paper showed that blood fat measurements at early stages of patients with RA are not useful for guiding methotrexate therapy, while clinical data alone gave better predictions” (51).
and (Benjamin Hur 2021) “Our study regarding plasma metabolites is consistent with another study in patients with RA with long-term disease duration (≥ 9 years) and using MTX among other biological medications. The study showed that certain metabolites, such as bilirubin and serine, were associated with low disease activity (54)”.
- In what way is this article different from already published articles of a similar nature.
https://pmc.ncbi.nlm.nih.gov/articles/PMC9404373/
Author reply: Thank you for your apprehension. To start with the published article is a systematic and scoping review, while ours is a systematic review. Our study applied all the guidelines for a systematic review and was registered in PROSPERO registry database to ensure transparency. Our study focused only on metabolites that predict the methotrexate treatment response, while the review mentions huge metabolites for MTX treatment response, but also for other therapies, for the diagnosis, and for the risk for development of RA disease...etc.
However, it is a good paper, and we have included it in our introduction (ref # 17). “Recently, it was shown that RA disease is accompanied by metabolic alterations resulting in huge metabolomic profiles. These metabolomic profiles can be determined using targeted and non-targeted metabolomics technology (17)”. Moreover, we have created a table (Table 1) comparing the two studies.
To highlight the unique contribution of this systematic review, Table 1 compares our study with existing systematic and scoping reviews, emphasizing its exclusive focus on methotrexate-specific predictive biomarkers in early rheumatoid arthritis (see Table [1]).
Table 1. Comparison of the Current Systematic Review with Existing Systematic and Scoping Reviews on Biomarkers in Rheumatoid Arthritis
|
Aspect |
Existing Systematic / Scoping Reviews |
Current Systematic review |
|
Main Focus |
Broadly evaluate metabolomics, clinical prediction models, or general biomarkers across various RA therapies (MTX, TNF inhibitors, IL-6 blockers, etc.). |
Focused exclusively on predictive biomarkers of methotrexate (MTX) treatment response in early rheumatoid arthritis (ERA). |
|
Objective |
Summarize associations between biomarkers and disease activity or treatment response, often across multiple drugs. |
Identify and validate metabolomic, proteomic, inflammatory, and immune-cell biomarkers predictive of MTX response for precision medicine. |
|
Study Type |
Systematic or scoping reviews, some with meta-analyses (e.g., MTX polyglutamates, clinical prediction models), heterogeneous in methods. |
Systematic review, PROSPERO-registered (CRD42024547651), PRISMA-compliant with clearly defined eligibility and synthesis methods. |
|
Data Scope |
Typically 10–25 studies including multiple RA therapies; focus on limited biomarker classes. |
16 MTX-specific human studies (2000–2024); 946 patients; 102 biomarkers analyzed; 31 biomarkers identified as functionally predictive. |
|
Therapeutic Scope |
Include multiple therapeutic agents; MTX findings are secondary or indirect. |
Dedicated entirely to MTX monotherapy in early RA. |
|
Omic Integration |
Usually single-omic focus (metabolomic or proteomic or genetic). |
Integrates multi-omic domains: metabolomic, proteomic, inflammatory, and immune-cell biomarkers. |
|
Clinical Relevance |
Rarely connect biomarkers with validated clinical indices (e.g., DAS28). |
Directly links biomarker patterns to DAS28-based clinical outcomes and treatment response categories. |
|
Novelty / Time Frame |
Mostly include studies up to 2023; limited integration of recent findings such as itaconate or IgG glycosylation. |
Incorporates the latest evidence through 2024, including novel predictors (itaconate, FA2-IgG1, RFC/fMRP, MTX-PG1–7). |
|
Outcome Orientation |
Emphasize biological plausibility and mechanistic interpretation. |
Provides a clinically actionable framework for individualized MTX therapy prediction. |
|
Scientific Contribution |
Offer general insights but lack MTX-specific predictive synthesis. |
Advances the field from descriptive multi-drug biomarker reviews to an MTX-specific, multi-omic predictive precision medicine model. |
- Based on the serum metabolites profile, why are three metabolites (N-methyl isoleucine, 2,3-dihydroxy butanoic acid, and nor-nicotine) associated with MTX therapy? What is the possible biochemical mechanism associated with it?
Author reply: Thank you for your question. Yes, in that specific study by Medcalf M R, et al. 2022, they found only 3 out of the 19 metabolites studied associated with good MTX-TR. Same goes for other studies that was why at the end we had 31 out of the total metabolites studied (102) considered as best predictive for MTX-TR. Of note the possible mechanisms associated with each metabolite are out of the scope of this study. Thus, we focused mainly on the clinical response to MTX therapy, however, further studies on the mechanisms of those 31 metabolites are needed.
- The authors must create a table providing the following information: upregulation/downregulation of serum lipids. Amino acids and other metabolites are in one category. For another column, they may use IgG, protein or other larger macromolecules.
Author reply: Thank you for your advice. We have created a table about upregulation and downregulation of the biomarkers.
Table 6 Categorized Summary of Predictive Biomarkers for Methotrexate Treatment Response in Early Rheumatoid Arthritis
|
Category |
Up regulated in Good Responders |
Downregulated in Good Responders |
Interpretation / Functional Notes |
|
Metabolites & Amino Acids |
Choline, Inosine, Hypoxanthine, Guanosine, Nicotinamide, Diglyceride |
N-methyl-isoleucine, 2,3-Dihydroxy-butanoic acid, Nor-nicotine, Glucosyl-ceramide, Itaconic acid |
Higher energy-related metabolites and purines indicate effective MTX metabolism and anti-inflammatory adenosine signaling; decreased branched-chain and lipid-derived acids associate with favorable response. |
|
Fatty-Acid / Lipid Pathway Metabolites |
Diglycerides, Nicotinamide-linked phospholipids |
Glucosylceramide, Itaconate pathway intermediates (itaconate, itaconate anhydrase, itaconate CoA) |
Altered lipid turnover reflects anti-inflammatory lipid remodeling; accumulation of itaconate derivatives predicts poor MTX response. |
|
Serum Proteins / Enzymes |
MRP8/14 complex (myeloid-related protein 8/14), Short-chain MTX-polyglutamates (MTX-PG1–7, esp. PG2), Functional multi-resistant protein (fMRP) + Reduced folate carrier (RFC) co-expression |
Haptoglobin (Hap) |
Elevated baseline inflammatory MRP8/14 and efficient intracellular MTX transport (fMRP + RFC, MTX-PGs) predict good response; high acute-phase haptoglobin indicates non-response. |
|
Immunoglobulin & Glycoprotein Markers |
— |
FA2 glycoform of IgG1 (low FA2/ (FA2G1 + FA2G2) ratio → non-response) |
Aberrant IgG1 Fc glycosylation linked to persistent inflammation and poor therapeutic outcome. |
|
Immune-Cell / Hematologic Parameters |
Higher baseline circulating transitional regulatory B cells (cTrB); Lower baseline Treg counts normalize post-therapy; Moderate rise in red-cell distribution width (RDW) after MTX initiation |
High baseline monocyte counts (CD14⁺high/low CD16⁻/⁺ subsets) |
cTrB enrichment and Treg restoration signify effective immune regulation; excess monocytes correlate with inadequate response. |
|
Cytokine / Inflammatory Ratios |
Lower IL-1ra / IL-1β ratio after MTX therapy |
High IL-6, CRP, CCL19 before treatment → risk of poor outcome |
Post-treatment IL-1β dominance indicates successful cytokine suppression cascade and therapeutic efficacy. |
Table 6 summarizes the 31 predictive biomarkers of methotrexate (MTX) treatment response in early rheumatoid arthritis (ERA), categorized into metabolites, lipids, serum proteins, immunoglobulins, immune cells, and cytokines. Upregulated biomarkers in good responders, such as choline, MTX-polyglutamates, and transitional regulatory B cells, reflect effective MTX metabolism and immune regulation, while downregulated markers like itaconate, haptoglobin, and high monocyte counts indicate poor response, linked to persistent inflammation. These findings provide a clinically actionable framework for predicting MTX efficacy, supporting personalized medicine in ERA.
- What do the authors mean by registered in PROSPERO in June 2024?
Author reply: Thank you for your apprehension, the authors need to stress the fact that the work started after the registration process was completed. However, we removed the sentence from the abstract but retained it in the manuscript for transparency. We also benefited from the information provided by the PROSPERO site that no similar study has been done before.

Reviewer 2 Report
Comments and Suggestions for Authors
This systematic review addresses an important and clinically relevant question regarding predictive biomarkers of methotrexate (MTX) response in early rheumatoid arthritis (RA). The paper is well-structured, includes a broad and recent literature search, and provides a valuable summary of 102 biomarkers, narrowing them to 31 with potential predictive value. The PROSPERO registration strengthens methodological transparency.
That said, several areas could be improved. First, while the introduction is comprehensive, some paragraphs are dense and could be streamlined to improve readability. The methods are generally clear, but the heterogeneity of included studies (different biomarkers, endpoints, and analytical techniques) limits comparability. Risk of bias (ROB) is mentioned but not systematically applied; a more consistent quality assessment would strengthen conclusions. Results are well presented in detailed tables and figures, but some are overly complex; clearer grouping or highlighting of key biomarkers would aid readers.
The discussion effectively synthesizes findings, but at times overstates clinical applicability. Given the variability of included studies, conclusions should emphasize that these biomarkers are promising but not yet ready for routine use. Finally, future directions could be more sharply defined, particularly regarding which biomarkers warrant validation in larger prospective trials.
Overall, this is a timely and valuable review but would benefit from revisions to improve clarity, consistency, and cautious interpretation.
Comments on the Quality of English LanguageThe manuscript is generally understandable and written in acceptable academic English. However, the text is sometimes wordy and repetitive, with long sentences that may reduce clarity. Certain sections, particularly the introduction and discussion, could be simplified and made more concise to improve readability. Minor grammatical issues, inconsistent use of tense, and occasional typographical errors are present but do not impede comprehension. Attention to sentence structure, smoother transitions between ideas, and careful proofreading would strengthen the overall presentation and ensure the paper reads more fluently for an international scientific audience.
Author Response
Reviewer 2
Comments and Suggestions for Authors
This systematic review addresses an important and clinically relevant question regarding predictive biomarkers of methotrexate (MTX) response in early rheumatoid arthritis (RA). The paper is well-structured, includes a broad and recent literature search, and provides a valuable summary of 102 biomarkers, narrowing them to 31 with potential predictive value. The PROSPERO registration strengthens methodological transparency.
That said, several areas could be improved.
First, while the introduction is comprehensive, some paragraphs are dense and could be streamlined to improve readability.
Author reply: Thank you for your concern, we agree with you. Accordingly, we revised the introduction, omitted some sentences, and highlighted the existing changes in yellow. We hope that improved the readability.
The methods are generally clear, but the heterogeneity of included studies (different biomarkers, endpoints, and analytical techniques) limits comparability.
Author reply: Thank you for your concern, we agree with your comment about heterogeneity of the studies included. To ensure streamlined flow to make this part clear, we moved some parts up and down. We also added to the method section some definitions about our research question using PICO framework, the primary effect measure used in our systematic review, and about the interpretation of the DAS28 score.
“Using the PICO framework our research question was as follows: In patients with early rheumatoid arthritis who are naïve to MTX therapy (P), how do baseline metabolomic biomarkers change after MTX treatment (I), compared to standard clinical measures alone (C), predict the therapeutic response to methotrexate (O), as measured by DAS28-ESR scores. The primary effect measure used in this study is the change in DAS28-ESR score from baseline following MTX treatment, that used to determine the clinical response (responder vs non-responder classification).”
Risk of bias (ROB) is mentioned but is not systematically applied; a more consistent quality assessment would strengthen conclusions.
Author reply: Thank you for your comments and advice. We have done the quality assessment using the Cochrane and PROBAST ROB tools and have added the following paragraph for the methods section:
“Among the 16 included studies, three were randomized clinical trials (RCTs) and thirteen were cohort studies. Accordingly, two tools were employed to assess the risk of bias (ROB): the Cochrane ROB tool for the randomized trials and the PROBAST tool for the cohort studies. Four reviewers independently assessed the ROB; disagreements were resolved by discussion with a fifth reviewer.
Regarding, Cochrane ROB tool (domains: random sequence generation, allocation concealment, blinding of participants/personnel, blinding of outcome assessment, incomplete outcome data, selective reporting, and other bias). Each domain was judged as “low”, “unclear” or “high” risk. An overall ROB judgment was derived as follows: studies were considered “low risk” if all domains were rated low, “high risk” if any domain was high, and “unclear” if no domain was high but at least one was unclear. Our results revealed that one study showed high ROB, and two studies showed unclear ROB.
On the other hand, the PROBAST framework evaluates ROB across four key domains: Participants, Predictors, Outcome, and Analysis. Following the PROBAST guidance, each domain was rated as 'Low,' 'High,' or 'Some Concerns,' with the overall ROB judgment determined by the highest risk rating across all domains. Out of the 13 studies, 11 studies (84.6%) were judged to have an Overall High ROB, while 2 studies (15.4%) were assessed as having Some Concerns regarding the overall ROB. The predominant source of bias was in the analysis domain, where most studies failed to perform appropriate statistical steps essential for prediction modeling, such as internal or external model validation.”
Results were in details explained as follows: “Based on the risk of bias assessment the included observational studies demonstrated variable methodological quality. The majority of studies (11 of 13) were rated as having an overall high risk of bias, primarily driven by concerns in the analysis domain, where issues such as inadequate confounder adjustment, missing data handling, or inappropriate statistical methods were identified. Two studies (Wolf et al., 2005 and Patro et al., 2016) achieved an overall rating of "some concerns," demonstrating better methodological rigor, particularly in their analytical approaches. Figure 5.
Based on the risk of bias assessment using the Cochrane Risk of Bias tool, the three included randomized controlled trials demonstrated moderate methodological quality with notable limitations. All three studies received an overall rating of either high or unclear risk of bias. The primary concerns stemmed from inadequate reporting of randomization procedures, with sequence generation being unclear in two studies (Fortea-Gordo P et al., 2020 and Hansen et al., 2006) and allocation concealment unclear in all three trials. Fortea-Gordo P et al. (2020) was rated as having high overall risk of bias due to lack of blinding of participants and personnel, which introduces potential performance bias. The remaining domains, including incomplete outcome data and selective reporting, were generally well-addressed across studies. The unclear risk of bias in key methodological domains, particularly regarding allocation concealment, limits confidence in the internal validity of these trials and suggests that the treatment effects may be subject to selection and performance bias. Figure 6.”
Results are well presented in detailed tables and figures, but some are overly complex; clearer grouping or highlighting of key biomarkers would aid readers.
Author reply: Thank you for your apprehension, we add more details in the results part about the dose used of MTX, interpretation of the DAS28 score and more explanation regarding Figure 2. Accordingly, we have added a paragraph to results section which we hope it makes our results clearer “Not all studies used the same dose of MTX. As shown in Table 3 most of the studied used high dose, 20 or 25 mg/week, but few studies used medium dose that ranged from 7.5 – 15 mg/ week. Of note, one important point when using the DAS28 score (0-10) to assess response to treatment, a DAS-28 reduction by 0.6 after therapy represents a moderate improvement, while a reduction of more than 1.2 represents a major improvement. Another important point regarding DAS28 is that a score of <2.6 suggests disease remission, 2.6-3.2 suggests low disease activity, >3.2-5.1 suggests moderate disease activity, while a score >5.1 suggests high disease activity. Figure 3 showed a major reduction of DAS28 score to low disease activity or even to remission state in most of the studies, which were considered major improvements. Figure 3 also showed that the disease activity got into remission (DAS28 <2.6) in 5 studies; Daly R. et al.2020 (Increased levels of itaconate and its derivatives in response to MTX treatment), Chara L, et al. 2015 (the higher pre-treatment number of circulating monocytes and their three subset cells), Nishina N, et al. 2013 (Serum IL-6 levels significantly reduced after MTX treatment in early RA pts), Wang, Z. et al. 2012 (Serum levels of 11 endogenous metabolites associated with good MTX-TR) and Wolf J, et al. 2004 (The absence or the presence of both functional multi resistant protein and reduced folate carrier (fMRP and RFC)”.
Therefore, maybe we should focus on those biomarkers reported by those 5 studies.
- Itaconate and its derivatives.
- circulating monocytes and their three subset cells.
- Serum IL-6 levels.
- Serum levels of 11 endogenous metabolites.
- Functional multi resistant protein and reduced folate carrier (fMRP and RFC).
The discussion effectively synthesizes findings, but at times overstates clinical applicability. Given the variability of included studies, conclusions should emphasize that these biomarkers are promising but not yet ready for routine use.
Author reply: Thank you for your comment, we agree with you and included your suggestions “these biomarkers are promising but not yet ready for routine use” in our conclusion and future direction. Also, we added that “We recommend that clinicians may focus on the biomarkers that got the patients into remission state as shown in Figure2. Those biomarkers as shown in Table 4 are: Itaconate and its derivatives, circulating monocytes and their three subset cells, serum IL-6 levels, serum levels of 11 endogenous metabolites, and functional multi resistant protein (fMRP), and reduced folate carrier (RFC).
Finally, future directions could be more sharply defined, particularly regarding which biomarkers warrant validation in larger prospective trials.
Overall, this is a timely and valuable review but would benefit from revisions to improve clarity, consistency, and cautious interpretation.
Author reply: Thank you for your vital comment, we agree with your opinion. After comprehensive revision to improve clarity, consistency, and cautious interpretation. We grouped the metabolites according to the reduction in the disease activity after MTX treatment into three groups. The three groups include those which resulted in: remission state, low disease activity, or moderate disease activity. Fortunately, no patients remained in high disease activity (DAS28>5.1) after treatment. Therefore, we recommend that these biomarkers, which resulted in the state of remission, are promising but not yet ready for routine clinical use. These biomarkers warrant validation in larger prospective trials. We also have created a figure (Figure 4), which could give a clear vision of our results.
Comments on the Quality of English Language
The manuscript is generally understandable and written in acceptable academic English. However, the text is sometimes wordy and repetitive, with long sentences that may reduce clarity. Certain sections, particularly the introduction and discussion, could be simplified and made more concise to improve readability. Minor grammatical issues, inconsistent use of tense, and occasional typographical errors are present but do not impede comprehension. Attention to sentence structure, smoother transitions between ideas, and careful proofreading would strengthen the overall presentation and ensure the paper reads more fluently for an international scientific audience.
Author reply: Thank you for your concern, we agree with you. Thus, we carefully revised the proofreading version, reduced long sentences, and simplified the introduction and the discussion to improve readability. Additionally, we requested the service for English editing by the Journal.

Reviewer 3 Report
Comments and Suggestions for Authors
While the topic of predictive biomarkers for methotrexate response in early rheumatoid arthritis is interesting, this systematic review has some flaws that prevent acceptance in the current form.
- Please use PICO framework explicitly to define the research question.
- What is you primary effect measure? Please explicitly state it in the methods and in the text
- The RoB assessement should have been more rigorous. The statement that RoB assessment is “helpful … but not necessary” is a methodological error. According to the Cochrane Handbook and PRISMA 2020 guidelines, assessment of RoB is a necessary component of any systematic review. As per supplementary materials you did RoB assessement but you should describe and discuss it in the paper and in the abstract, preferably you should present some graphs. Without clear explicit information about RoB in all studies the paper is not understandable for readers.
- "Methods section" is too concise. Please write in more detail including RoB assessement with reference, statistical analyses etc.
- The conclusion seems to be overstated. There is a claim to provide guidance for precision medicine, but it is no supported by the evidence presented.
Author Response
Reviewer 3
Comments and Suggestions for Authors
While the topic of predictive biomarkers for methotrexate response in early rheumatoid arthritis is interesting, this systematic review has some flaws that prevent acceptance in the current form.
- Please use PICO framework explicitly to define the research question.
Author reply: Thank you for your apprehension. We have added to the method section the following paragraph about PICO framework “We chose DAS28-ESR score as it was proved to be more effective in assessing MTX response in ERA patients than other measurement scores or tools. Accordingly, our research question using the PICO framework as follows: In patients with early rheumatoid arthritis who are naïve to MTX therapy (P), how do baseline metabolomic biomarkers change after MTX treatment (I), compared to standard clinical measures alone (C), predict the therapeutic response to methotrexate (O), as measured by DAS28-ESR scores”.
- What is your primary effect measure? Please explicitly state it in the methods and in the text.
Author reply: Thank you for your concern. We have added to the method section the following paragraph “The primary effect measure used in this study is the change in DAS28-ESR score from baseline following MTX treatment, used to determine clinical response (responder vs. non-responder classification)”.
- The RoB assessment should have been more rigorous. The statement that RoB assessment is “helpful … but not necessary” is a methodological error. According to the Cochrane Handbook and PRISMA 2020 guidelines, assessment of RoB is a necessary component of any systematic review. As per supplementary materials you did RoB assessment, but you should describe and discuss it in the paper and in the abstract, preferably you should present some graphs. Without clear explicit information about RoB in all studies the paper is not understandable for readers.
Author reply: Thank you for your comments and advice. We have done quality assessment using the Cochrane and PROBAST ROB tools and have added a paragraph to the methods section as explained in question #4 below.
- "Methods section" is too concise. Please write in more detail including RoB assessment with reference, statistical analyses etc.
Author reply: Thank you for your comments and advice. We have done quality assessment using the Cochrane and PROBAST ROB tools and have added the following paragraph for the methods section:
“Among the 16 included studies, three were randomized clinical trials (RCTs) and thirteen were cohort studies. Accordingly, two tools were employed to assess the risk of bias (ROB): the Cochrane ROB tool for the randomized trials and the PROBAST tool for the cohort studies. Four reviewers independently assessed the ROB; disagreements were resolved by discussion with a fifth reviewer.
Regarding, Cochrane ROB tool (domains: random sequence generation, allocation concealment, blinding of participants/personnel, blinding of outcome assessment, incomplete outcome data, selective reporting, and other bias). Each domain was judged as “low”, “unclear” or “high” risk. An overall ROB judgment was derived as follows: studies were considered “low risk” if all domains were rated low, “high risk” if any domain was high, and “unclear” if no domain was high but at least one was unclear. Our results revealed that one study showed high ROB, and two studies showed unclear ROB.
On the other hand, the PROBAST framework evaluates ROB across four key domains: Participants, Predictors, Outcome, and Analysis. Following the PROBAST guidance, each domain was rated as 'Low,' 'High,' or 'Some Concerns,' with the overall ROB judgment determined by the highest risk rating across all domains. Out of the 13 studies, 11 studies (84.6%) were judged to have an Overall High ROB, while 2 studies (15.4%) were assessed as having Some Concerns regarding the overall ROB. The predominant source of bias was in the analysis domain, where most studies failed to perform appropriate statistical steps essential for prediction modeling, such as internal or external model validation.”
- The conclusion seems to be overstated. There is a claim to provide guidance for precision medicine, but it is not supported by the evidence presented.
Author reply: Thank you for your comment. We have added to paragraph and joined it with the conclusion, and also a future direction part “Furthermore, we revealed that few biomarkers (out of the 31) resulted in state of remission. These biomarkers are: Itaconate and its derivatives, circulating monocytes and their three subset cells, serum IL-6 levels, serum levels of 11 endogenous metabolites, and functional multi resistant protein (fMRP) together with reduced folate carrier (RFC). These biomarkers are promising but not yet ready for routine use. We recommend that clinicians should focus on these biomarkers that got the patients into remission state as first line biomarkers to be used for routine clinical practice. Further studies are still needed to identify subgroups or clinical RA phenotypes that will respond to MTX treatment or other treatment strategies. However, the implementation of personalized medicine in early RA patients still re-quires further validation in larger prospective trials of those 31 biomarkers provided by our study and combining them not only with baseline DAS28 scores, but also with clinical phenotypes to find out which ones are applicable for the daily clinical practice, for accurate prediction of MTX treatment response in early RA based on individual patient characteristics”.
Future directions
We revealed that few biomarkers resulted in state of remission in patients with ERA. These biomarkers are promising but not yet ready for routine clinical use, they warrant validation in larger prospective trials. We recommend that for the implementation of personalized medicine these biomarkers should be the first line biomarkers to be used for routine clinical practice after validation.

Round 2
Reviewer 1 Report
Comments and Suggestions for Authors
The authors made a significant change in the manuscript, and it looks much better now. The authors can be congratulated for their extensive efforts.
Comments on the Quality of English LanguageNone